# DAB-DETR: Dynamic Anchor Boxes are Better Queries for DETR

**Shilong Liu**[1,2*], **Feng Li**[2,3], **Hao Zhang**[2,3], **Xiao Yang**[1],
**Xianbiao Qi**[2], **Hang Su**[1,4], **Jun Zhu**[1,4†], **Lei Zhang**[2†]

[1]Dept. of Comp. Sci. and Tech., BNRist Center, State Key Lab for Intell. Tech.
& Sys., Institute for AI, Tsinghua-Bosch Joint Center for ML, Tsinghua University.
[2]International Digital Economy Academy (IDEA).
[3]Hong Kong University of Science and Technology.
[4]Peng Cheng Laboratory, Shenzhen, Guangdong, China.
{liusl20,yangxiao19}@mails.tsinghua.edu.cn
{fliay,hzhangcx}@connect.ust.hk
{qixianbiao,leizhang}@idea.edu.cn
{suhangss,dcszj}@mail.tsinghua.edu.cn

## Abstract

We present in this paper a novel query formulation using dynamic anchor boxes for DETR (DEtection TRansformer) and offer a deeper understanding of the role of queries in DETR. This new formulation directly uses box coordinates as queries in Transformer decoders and dynamically updates them layer by layer. Using box coordinates not only helps using explicit positional priors to improve the query-to-feature similarity and eliminate the slow training convergence issue in DETR, but also allows us to modulate the positional attention map using the box width and height information. Such a design makes it clear that queries in DETR can be implemented as performing soft ROI pooling layer by layer in a cascade manner. As a result, it leads to the best performance on MS-COCO benchmark among the DETR-like detection models under the same setting, e.g., AP 45.7% using ResNet50-DC5 as backbone trained in 50 epochs. We also conducted extensive experiments to confirm our analysis and verify the effectiveness of our methods. Code is available at https://github.com/IDEA-opensource/DAB-DETR.

## 1 Introduction

Object detection is a fundamental task in computer vision of wide applications. Most classical detectors are based on convolutional architectures which have made remarkable progress in the last decade (Ren et al., 2017; Girshick, 2015; Redmon et al., 2016; Bochkovskiy et al., 2020; Ge et al., 2021). Recently, Carion et al. (2020) proposed a Transformer-based end-to-end detector named DETR (DEtection TRansformer), which eliminates the need for hand-designed components, e.g., anchors, and shows promising performance compared with modern anchor-based detectors such as Faster RCNN (Ren et al., 2017).

In contrast to anchor-based detectors, DETR models object detection as a set prediction problem and uses 100 learnable queries to probe and pool features from images, which makes predictions without the need of using non-maximum suppression. However, due to its ineffective design and use of queries, DETR suffers from significantly slow training convergence, usually requiring 500 epochs to achieve a good performance. To address this issue, many follow-up works attempted to improve the design of DETR queries for both faster training convergence and better performance (Zhu et al., 2021; Gao et al., 2021; Meng et al., 2021; Wang et al., 2021).

---

*This work was done when Shilong Liu was intern at IDEA.
†Corresponding author.

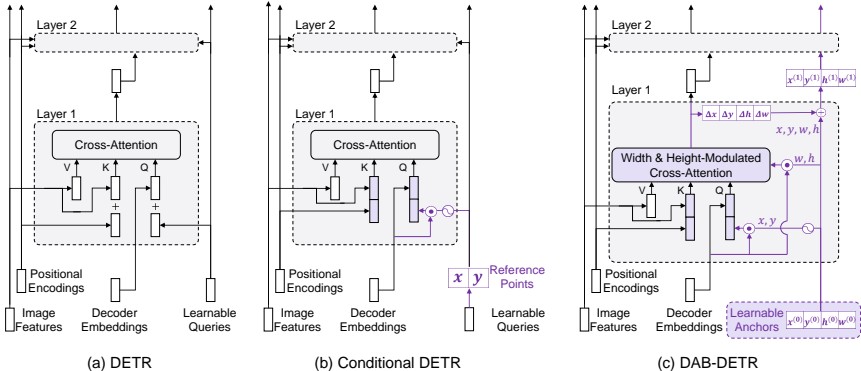

Figure 1: Comparison of DETR, Conditional DETR, and our proposed DAB-DETR. For clarity, we only show the cross-attention part in the Transformer decoder. (a) DETR uses the learnable queries for all the layers without any adaptation, which accounts for its slow training convergence. (b) Conditional DETR adapts the learnable queries for each layer mainly to provide a better reference query point to pool features from the image feature map. In contrast, (c) DAB-DETR directly uses dynamically updated anchor boxes to provide both a reference query point $(x, y)$ and a reference anchor size $(w, h)$ to improve the cross-attention computation. We marked the modules with difference in purple.

Despite all the progress, the role of the learned queries in DETR is still not fully understood or utilized. While most previous attempts make each query in DETR more explicitly associated with one specific spatial position rather than multiple positions , the technical solutions are largely different. For example, Conditional DETR learns a conditional spatial query by adapting a query based on its content feature for better matching with image features (Meng et al., 2021). Efficient DETR introduces a dense prediction module to select top-K object queries (Yao et al., 2021) and Anchor DETR formulates queries as 2D anchor points (Wang et al., 2021), both associating each query with a specific spatial position. Similarly, Deformable DETR directly treats 2D reference points as queries and performs deformable cross-attention operation at each reference points (Zhu et al., 2021). But all the above works only leverage 2D positions as anchor points without considering the object scales.

Motivated by these studies, we take a closer look at the cross-attention module in Transformer decoder and propose to use anchor boxes, i.e., 4D box coordinates $(x, y, w, h)$, as queries in DETR and update them layer by layer. This new query formulation introduce better spatial priors for the cross-attention module by considering both the position and size of each anchor box, which also leads to a much simpler implementation and a deeper understanding of the role of queries in DETR.

The key insight behind this formulation is that each query in DETR is formed by two parts: a content part (decoder self-attention output) and a positional part (e.g., learnable queries in DETR) [1]. The cross-attention weights are computed by comparing a query with a set of keys which consists of two parts as a content part (encoded image feature) and a positional part (positional embedding). Thus, queries in Transformer decoder can be interpreted as pooling features from a feature map based on the query-to-feature similarity measure, which considers both the content and positional information. While the content similarity is for pooling semantically related features, the positional similarity is to provide a positional constraint for pooling features around the query position. This attention computing mechanism motivates us to formulate queries as anchor boxes as illustrated in Fig. 1 (c), allowing us to use the center position $(x, y)$ of an anchor box to pool features around the center and use the anchor box size $(w, h)$ to modulate the cross-attention map, adapting it to anchor box size. In addition, because of the use of coordinates as queries, anchor boxes can be updated layer by layer dynamically. In this way, queries in DETR can be implemented as performing soft ROI pooling layer by layer in a cascade way.

We provide a better positional prior for pooling features by using anchor box size to modulate the cross-attention. Because the cross-attention can pool features from the whole feature map, it is

---

[1]See the DETR implementation at https://github.com/facebookresearch/detr. The components of queries and keys are also shown in each subplot of Fig. 1. Note that the learnable queries in DETR are only for the positional part. Related discussion can also be found in Conditional DETR (Meng et al., 2021).

crucial to provide a proper positional prior for each query to let the cross-attention module focus on a local region corresponding to a target object. It can also facilitate to speed up the training convergence of DETR. Most prior works improve DETR by associating each query with a specific location, but they assume an isotropic Gaussian positional prior of a fixed size(Fig. 4 (b)), which is inappropriate for objects of different scales. With the size information $(w, h)$ available in each query anchor box, we can modulate the Gaussian positional prior as an oval shape. More specifically, we divide the width and height from the cross-attention weight (before softmax) for its $x$ part and $y$ part separately, which helps the Gaussian prior to better match with objects of different scales(Fig. 4 (c)). To further improve the positional prior, we also introduce a temperature parameter to tune the flatness of positional attention, which has been overlooked in all prior works.

In summary, our proposed DAB-DETR (**D**ynamic **A**nchor **B**ox **DETR**) presents a novel query formulation by directly learning anchors as queries. This formulation offers a deeper understanding of the role of queries, allowing us to use anchor size to modulate the positional cross-attention map in Transformer decoders and perform dynamic anchor update layer by layer. Our results demonstrate that DAB-DETR attains the best performance among DETR-like architectures under the same setting on the COCO object detection benchmark. The proposed method can achieve $45.7\%$ AP when using a single ResNet-50 (He et al., 2016) model as backbone for training 50 epochs. We also conducted extensive experiments to confirm our analysis and verify the effectiveness of our methods.

## 2 RELATED WORK

Most classical detectors are anchor-based, using either anchor boxes (Ren et al., 2017; Girshick, 2015; Sun et al., 2021) or anchor points (Tian et al., 2019; Zhou et al., 2019). In contrast, DETR (Carion et al., 2020) is a fully anchor-free detector using a set of learnable vectors as queries. Many follow-up works attempted to solve the slow convergence of DETR from different perspectives. Sun et al. (2020) pointed out that the cause of slow training of DETR is due to the cross-attention in decoders and hence proposed an encoder-only model. Gao et al. (2021) instead introduced a Gaussian prior to regulate the cross-attention. Despite their improved performance, they did not give a proper explanation of the slow training and the roles of queries in DETR.

Another direction to improve DETR, which is more relevant to our work, is towards a deeper understanding of the role of queries in DETR. As the learnable queries in DETR are used to provide positional constraints for feature pooling, most related works attempted to make each query in DETR more explicitly related to a specific spatial position rather than multiple position modes in the vanilla DETR. For example, Deformable DETR (Zhu et al., 2021) directly treats 2D reference points as queries and predicts deformable sampling points for each reference point to perform the deformable cross-attention operation. Conditional DETR (Meng et al., 2021) decouples the attention formulation and generates positional queries based on reference coordinates. Efficient DETR (Yao et al., 2021) introduces a dense prediction module to select top-K positions as object queries. Although these works connect queries with positional information, they do not have an explicit formulation to use anchors.

Different from the hypothesis in prior works that the learnable query vectors contain box coordinate information, our approach is based on a new perspective that all information contained in queries are box coordinates. That is, *anchor boxes are better queries for DETR*. A concurrent work Anchor DETR (Wang et al., 2021) also suggests learning anchor points directly, while it ignores the anchor width and height information as in other prior works. Besides DETR, Sun et al. (2021) proposed a sparse detector by learning boxes directly, which shares a similar anchor formulation with us, but it discards the Transformer structure and leverages hard ROI align for feature extraction. Table 1 summarizes the key differences between related works and our proposed DAB-DETR. We compare our model with related works on five dimensions: if the model directly learns anchors, if the model predicts reference coordinates (in its intermediate stage), if the model updates the reference anchors layer by layer, if the model uses the standard dense cross-attention, if the attention is modulated to better match with objects of different scales, and if the model updates the learned queries layer by layer. A more detailed comparison of DETR-like models is available in Sec. B of Appendix. We recommend this section for readers who have confusions about the table.

| Models | Learn Anchors? | Reference Anchors | Dynamic Anchors | Standard Attention | Size-Modulated Attention | Update Learned Spatial Queries? |
|---|---|---|---|---|---|---|
| DETR | No | No | | ✓ | | |
| Deformable DETR | No | 4D | ✓ | | ✓ | |
| SMCA | No | 4D | ✓ | | ✓ | |
| Conditional DETR | No | 2D | | ✓ | | |
| Anchor DETR | 2D | 2D | ✓ | | | |
| Sparse RCNN | 4D | 4D | ✓ | | | |
| DAB-DETR | 4D | 4D | ✓ | ✓ | ✓ | ✓ |

Table 1: Comparison of representative related models and our DAB-DETR. The term "Learn Anchors?" asks if the model learns 2D points or 4D anchors as learnable parameters directly. The term "Reference Anchors" means if the model predicts relative coordinates with respect to a reference points/anchors. The term "Dynamic Anchors" indicates if the model updates its anchors layer-by-layer. The term "Standard Attention" shows whether the model leverages the standard dense attention in cross-attention modules. The term "Object Scale-Modulated Attention" means if the attention is modulated to better match with object scales. The term "Size-Modulated Attention" means if the attention is modulated to better match with object scales. The term "Update Spatial Learned Queries?" means if the learned queries are updated layer by layer. Note that Sparse RCNN is not a DETR-like architecture. we list it here for their similar anchor formulation with us. See Sec. B of Appendix for a more detailed comparison of these models.

## 3 WHY A POSITIONAL PRIOR COULD SPEEDUP TRAINING?

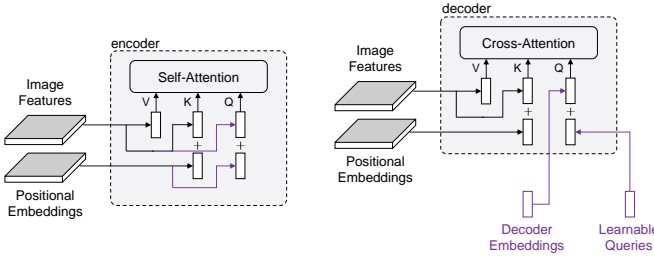

(a) Self-attention in encoder of DETR    (b) Cross-attention in decoder of DETR

Figure 2: Comparison of self-attention in encoders and cross-attention in decoders of DETR. As they have the same key and value components, the only difference comes from the queries. Each query in an encoder is composed of an image feature (content information) and a positional embedding (positional information), whereas each query in a decoder is composed of a decoder embedding (content information) and a learnable query (postional information). The differences between two modules are marked in purple.

Much work has been done to accelerate the training convergence speed of DETR, while lacking a unified understanding of why their methods work. Sun et al. (2020) showed that the cross-attention module is mainly responsible for the slow convergence, but they simply removed the decoders for faster training. We follow their analysis to find which sub-module in the cross-attention affects the performance. Comparing the self-attention module in encoders with the cross-attention module in decoders, we find the key difference between their inputs comes from the queries, as shown in Fig. 2. As the decoder embeddings are initialized as 0, they are projected to the same space as the image features after the first cross-attention module. After that, they will go through a similar process in decoder layers as the image features in encoder layers. Hence the root cause is likely due to the learnable queries.

Two possible reasons in cross-attention account for the model's slow training convergence: 1) it is hard to learn the queries due to the optimization challenge, and 2) the positional information in the learned queries is not encoded in the same way as the sinusoidal positional encoding used for image features. To see if it is the first reason, we reuse the well-learned queries from DETR (keep them fixed) and only train the other modules. The training curves in Fig. 3(a) show that the fixed queries only slightly improve the convergence in very early epochs, e.g., the first 25 epochs. Hence the query learning (or optimization) is likely not the key concern.

Then we turn to the second possibility and try to find out if the learned queries have some undesirable properties. As the learned queries are used to filter objects in certain regions, we visualize a few positional attention maps between the learned queries and the positional embeddings of image

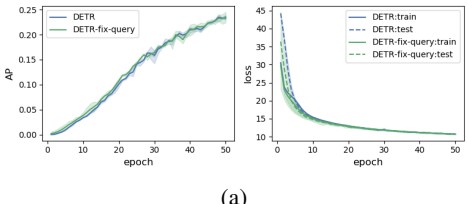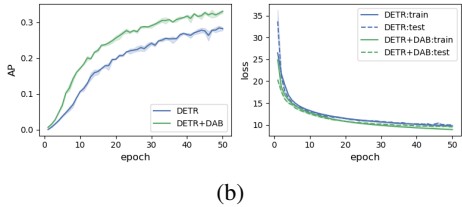

(a)                                                                          (b)

Figure 3: a): Training curves of the original DETR and DETR with fixed queries. b): Training curves of the original DETR and DETR+DAB. We run each experiment 3 times and plot the mean value and the 95% confidence interval of each item.

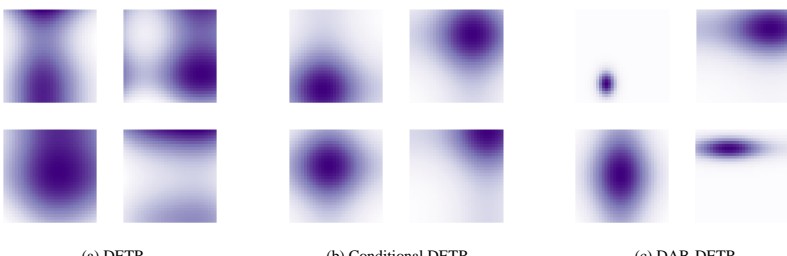

(a) DETR                    (b) Conditional DETR                    (c) DAB-DETR

Figure 4: We visualize the positional attention between positional queries and positional keys for DETR, Conditional DETR, and our proposed DAB-DETR. Four attention maps in (a) are randomly sampled, and we select figures with similar query positions as in (a) for (b) and (c). The darker the color, the greater the attention weight, and vice versa. (a) Each attention map in DETR is calculated by performing dot product between a learned query and positional embeddings from a feature map, and can have multiple modes and unconcentrated attentions. (b) The positional queries in Conditional DETR are encoded in the same way as the image positional embeddings, resulting in Gaussian-like attention maps. However, it cannot adapt to objects of different scales. (c) DAB-DETR explicitly modulates the attention map using the width and height information of an anchor, making it more adaptive to object size and shape. The modulated attentions can be regarded as helping perform soft ROI pooling.

features in Fig. 4(a). Each query can be regarded as a positional prior to let decoders focus on a region of interest. Although they serve as a positional constraint, they also carry undesirable properties: multiple modes and nearly uniform attention weights. For example, the two attention maps at the top of Fig. 4(a) have two or more concentration centers, making it hard to locate objects when multiple objects exist in an image. The bottom maps of Fig. 4(a) focus on areas that are either too large or too small, and hence cannot inject useful positional information into the procedure of feature extraction. We conjecture that the multiple mode property of queries in DETR is likely the root cause for its slow training and we believe introducing explicit positional priors to constrain queries on a local region is desirable for training. To verify this assumption, we replace the query formulation in DETR with dynamic anchor boxes, which can enforce each query to focus on a specific area, and name this model DETR+DAB. The training curves in Fig. 3(b) show that DETR+DAB leads to much better performance compared with DETR, in terms of both detection AP and training/testing loss. Note that the only difference between DETR and DETR+DAB is the formulation of queries and no other techniques like 300 queries or focal loss are introduced. It shows that after addressing the multi-mode issue of DETR queries, we can achieve both a faster training convergence and a higher detection accuracy.

Some previous works also have similar analyses and confirmed this. For example, SMCA (Gao et al., 2021) speeds up the training by applying pre-defined Gaussian maps around reference points. Conditional DETR (Meng et al., 2021) uses explicit positional embedding as positional queries for training, yielding attention maps similar to Gaussian kernels as shown in Fig. 4(b). Although explicit positional priors lead to good performance in training, they ignore the scale information of an object. In contrast, our proposed DAB-DETR explicitly takes into account the object scale information to adaptively adjust attention weights, as shown in Fig. 4(c).

# 4   DAB-DETR

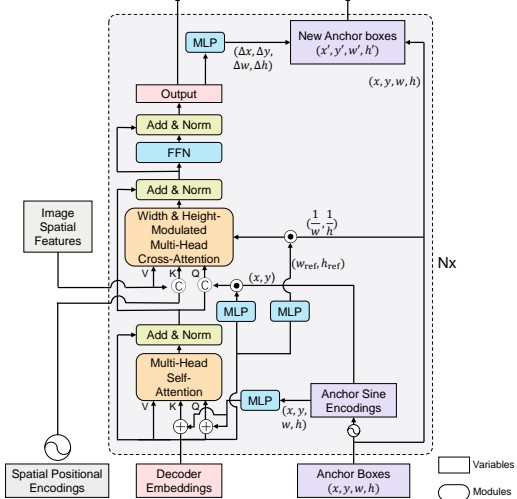

Figure 5: Framework of our proposed DAB-DETR.

## 4.1   Overview

Following DETR (Carion et al., 2020), our model is an end-to-end object detector which includes a CNN backbone, Transformer (Vaswani et al., 2017) encoders and decoders, and prediction heads for boxes and labels. We mainly improve the decoder part, as shown in Fig. 5.

Given an image, we extract image spatial features using a CNN backbone followed with Transformer encoders to refine the CNN features. Then dual queries, including positional queries (anchor boxes) and content queries (decoder embeddings), are fed into the decoder to probe the objects which correspond to the anchors and have similar patterns with the content queries. The dual queries are updated layer by layer to get close to the target ground-truth objects gradually. The outputs of the final decoder layer are used to predict the objects with labels and boxes by prediction heads, and then a bipartite graph matching is conducted to calculate loss as in DETR.

To illustrate the generality of our dynamic anchor boxes, we also design a stronger DAB-Deformable-DETR, which is available in Appendix.

## 4.2   Learning Anchor Boxes Directly

As discussed in Sec. 1 regarding the role of queries in DETR, we propose to directly learn query boxes or say anchor boxes and derive positional queries from these anchors. There are two attention modules in each decoder layer, including a self-attention module and a cross-attention module, which are used for query updating and feature probing, respectively. Each module needs queries, keys, and values to perform attention-based value aggregation, yet the inputs of these triplets differ.

We denote $A_q = (x_q, y_q, w_q, h_q)$ as the $q$-th anchor, $x_q, y_q, w_q, h_q \in \mathbb{R}$, and $C_q \in \mathbb{R}^D$ and $P_q \in \mathbb{R}^D$ as its corresponding content query and positional query, where $D$ is the dimension of decoder embeddings and positional queries.

Given an anchor $A_q$, its positional query $P_q$ is generated by:

$$P_q = \text{MLP}(\text{PE}(A_q)), \tag{1}$$

where PE means positional encoding to generate sinusoidal embeddings from float numbers and the parameters of MLP are shared across all layers. As $A_q$ is a quaternion, we overload the PE operator here:

$$\text{PE}(A_q) = \text{PE}(x_q, y_q, w_q, h_q) = \text{Cat}(\text{PE}(x_q), \text{PE}(y_q), \text{PE}(w_q), \text{PE}(h_q)). \tag{2}$$

The notion Cat means concatenation function. In our implementations, the positional encoding function PE maps a float to a vector with $D/2$ dimensions as: PE: $\mathbb{R} \to \mathbb{R}^{D/2}$. Hence the function MLP projects a $2D$ dimensional vector into $D$ dimensions: MLP: $\mathbb{R}^{2D} \to \mathbb{R}^D$. The MLP module has two submodules, each of which is composed of a linear layer and a ReLU activation, and the feature reduction is conducted at the first linear layer.

In the self-attention module, all three of queries, keys, and values have the same content items, while the queries and keys contain extra position items:

$$\text{Self-Attn:} \quad Q_q = C_q + P_q, \quad K_q = C_q + P_q, \quad V_q = C_q, \tag{3}$$

Inspired by Conditional DETR (Meng et al., 2021), we concatenate the position and content information together as queries and keys in the cross-attention module, so that we can decouple the content and position contributions to the query-to-feature similarity computed as the dot product between a query and a key. To rescale the positional embeddings, we leverage the conditional spatial query (Meng et al., 2021) as well. More specifically, we learn a $\text{MLP}^{(\text{csq})} : \mathbb{R}^D \to \mathbb{R}^D$ to obtain a scale vector conditional on the content information and use it perform element-wise multiplication with the positional embeddings:

$$\begin{aligned} \text{Cross-Attn:} \quad Q_q &= \text{Cat}(C_q, \text{PE}(x_q, y_q) \cdot \text{MLP}^{(\text{csq})}(C_q)), \\ K_{x,y} &= \text{Cat}(F_{x,y}, \text{PE}(x, y)), \quad V_{x,y} = F_{x,y}, \end{aligned} \tag{4}$$

where $F_{x,y} \in \mathbb{R}^D$ is the image feature at position $(x, y)$ and $\cdot$ is an element-wise multiplication. Both the positional embeddings in queries and keys are generated based on 2D coordinates, making it more consistent to compare the positional similarity, as in previous works (Meng et al., 2021; Wang et al., 2021).

### 4.3 ANCHOR UPDATE

Using coordinates as queries for learning makes it possible to update them layer by layer. In contrast, for queries of high dimensional embeddings, such as in DETR (Carion et al., 2020) and Conditional DETR (Meng et al., 2021), it is hard to perform layer-by-layer query refinement, because it is unclear how to convert an updated anchor back to a high-dimensional query embedding.

Following the previous practice (Zhu et al., 2021; Wang et al., 2021), we update anchors in each layer after predicting relative positions $(\Delta x, \Delta y, \Delta w, \Delta h)$ by a prediction head, as shown in Fig. 5. Note that all prediction heads in different layers share the same parameters.

### 4.4 WIDTH & HEIGHT-MODULATED GAUSSIAN KERNEL

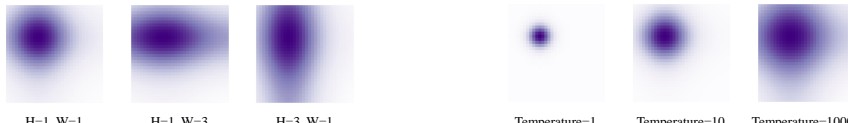

H=1, W=1    H=1, W=3    H=3, W=1          Temperature=1   Temperature=10   Temperature=10000

Figure 6: Positional attention maps modulated by width and height.

Figure 7: Positional attention maps with different temperatures.

Traditional positional attention maps are used as a Gaussian-like prior, as shown in Fig. 6 left. But the prior is simply assumed isotropic and fixed size for all objects, leaving their scale information (width and height) ignored. To improve the positional prior, we propose to inject the scale information into the attention maps.

The query-to-key similarity in the original positional attention map is computed as the sum of dot products of two coordinate encodings:

$$\text{Attn}((x, y), (x_{\text{ref}}, y_{\text{ref}})) = (\text{PE}(x) \cdot \text{PE}(x_{\text{ref}}) + \text{PE}(y) \cdot \text{PE}(y_{\text{ref}}))/\sqrt{D}, \tag{5}$$

where $1/\sqrt{D}$ is used to rescale the value as suggested in Vaswani et al. (2017). We modulate the positional attention maps (before softmax) by dividing the relative anchor width and height from its $x$ part and $y$ part separately to smooth the Gaussian prior to better match with objects of different scales:

$$\text{ModulateAttn}((x, y), (x_{\text{ref}}, y_{\text{ref}})) = (\text{PE}(x) \cdot \text{PE}(x_{\text{ref}})\frac{w_{q,\text{ref}}}{w_q} + \text{PE}(y) \cdot \text{PE}(y_{\text{ref}})\frac{h_{q,\text{ref}}}{h_q})/\sqrt{D}, \tag{6}$$

where $w_q$ and $h_q$ are the width and height of the anchor $A_q$, and $w_{q,\text{ref}}$ and $h_{q,\text{ref}}$ are the reference width and height that are calculated by:

$$w_{q,\text{ref}}, h_{q,\text{ref}} = \sigma(\text{MLP}(C_q)). \tag{7}$$

This modulated positional attention helps us extract features of objects with different widths and heights, and the visualizations of modulated attentions are shown in Fig. 6.

### 4.5 TEMPERATURE TUNING

For position encoding, we use the sinusoidal function (Vaswani et al., 2017), which is defined as:

$$\text{PE}(x)_{2i} = \sin(\frac{x}{T^{2i/D}}), \quad \text{PE}(x)_{2i+1} = \cos(\frac{x}{T^{2i/D}}), \tag{8}$$

where $T$ is a hand-design temperature, and the superscript $2i$ and $2i + 1$ denote the indices in the encoded vectors. The temperature $T$ in Eq. (8) influences the size of positional priors, as shown in Fig. 7. A larger $T$ results in a more flattened attention map, and vice versa. Note that the temperature $T$ is hard-coded in (Vaswani et al., 2017) as 10000 for natural language processing, in which the values of $x$ are integers representing each word's position in a sentence. However, in DETR, the values of $x$ are floats between 0 and 1 representing bounding box coordinates. Hence a different temperature is highly desired for vision tasks. In this work, we empirically choose $T = 20$ in all our models.

## 5 EXPERIMENTS

We provide the training details in Appendix A.

### 5.1 MAIN RESULTS

Table 2 shows our main results on the COCO 2017 validation set. We compare our proposed DAB-DETR with DETR (Carion et al., 2020), Faster RCNN (Ren et al., 2017), Anchor DETR (Wang et al., 2021), SMCA (Gao et al., 2021), Deformable DETR (Zhu et al., 2021), TSP (Sun et al., 2020), and Conditional DETR (Meng et al., 2021). We showed two variations of our model: standard models and models marked with superscript * that have 3 pattern embeddings (Wang et al., 2021). Our standard models outperform Conditional DETR with a large margin. We notice that our model introduces a slight increase of GFLOPs. GFLOPs may differ depending on the calculation scripts and we use the results reported by the authors in Table 2. Actually, we find in our tests that the GFLOPs of our standard models are nearly the same as the corresponding Conditional DETR models based on our GFLOPs calculation scripts, hence our model still has advantages over previous work under the same settings. When using pattern embeddings, our DAB-DETR with * outperforms previous DETR-like methods on all four backbones with a large margin, even better than multi-scale architectures. It verifies the correctness of our analysis and the effectiveness of our design.

### 5.2 ABLATIONS

Table 3 shows the effectiveness of each component in our model. We find that all modules we proposed contribute remarkably to our final results. The anchor box formulation improves the performance from 44.0% AP to 45.0% AP compared with the anchor point formulation (compare Row

| Model | MultiScale | #epochs | AP | $AP_{50}$ | $AP_{75}$ | $AP_S$ | $AP_M$ | $AP_L$ | GFLOPs | Params |
|---|---|---|---|---|---|---|---|---|---|---|
| DETR-R50 | | 500 | 42.0 | 62.4 | 44.2 | 20.5 | 45.8 | 61.1 | 86 | 41M |
| Faster RCNN-FPN-R50 | | 108 | 42.0 | 62.1 | 45.5 | 26.6 | 45.5 | 53.4 | 180 | 42M |
| Anchor DETR-R50* | | 50 | 42.1 | 63.1 | 44.9 | 22.3 | 46.2 | 60.0 | – | 39M |
| Conditional DETR-R50 | | 50 | 40.9 | 61.8 | 43.3 | 20.8 | 44.6 | 59.2 | 90 | 44M |
| DAB-DETR-R50 | | 50 | 42.2 | 63.1 | 44.7 | 21.5 | 45.7 | 60.3 | 94 | 44M |
| DAB-DETR-R50* | | 50 | **42.6** | 63.2 | 45.6 | 21.8 | 46.2 | 61.1 | 100 | 44M |
| DETR-DC5-R50 | | 500 | 43.3 | 63.1 | 45.9 | 22.5 | 47.3 | 61.1 | 187 | 41M |
| Deformable DETR-R50 | ✓ | 50 | 43.8 | 62.6 | 47.7 | 26.4 | 47.1 | 58.0 | 173 | 40M |
| SMCA-R50 | ✓ | 50 | 43.7 | 63.6 | 47.2 | 24.2 | 47.0 | 60.4 | 152 | 40M |
| TSP-RCNN-R50 | ✓ | 96 | 45.0 | 64.5 | 49.6 | 29.7 | 47.7 | 58.0 | 188 | – |
| Anchor DETR-DC5-R50* | | 50 | 44.2 | 64.7 | 47.5 | 24.7 | 48.2 | 60.6 | 151 | 39M |
| Conditional DETR-DC5-R50 | | 50 | 43.8 | 64.4 | 46.7 | 24.0 | 47.6 | 60.7 | 195 | 44M |
| DAB-DETR-DC5-R50 | | 50 | 44.5 | 65.1 | 47.7 | 25.3 | 48.2 | 62.3 | 202 | 44M |
| DAB-DETR-DC5-R50* | | 50 | **45.7** | 66.2 | 49.0 | 26.1 | 49.4 | 63.1 | 216 | 44M |
| DETR-R101 | | 500 | 43.5 | 63.8 | 46.4 | 21.9 | 48.0 | 61.8 | 152 | 60M |
| Faster RCNN-FPN-R101 | | 108 | 44.0 | 63.9 | 47.8 | 27.2 | 48.1 | 56.0 | 246 | 60M |
| Anchor DETR-R101* | | 50 | 43.5 | 64.3 | 46.6 | 23.2 | 47.7 | 61.4 | – | 58M |
| Conditional DETR-R101 | | 50 | 42.8 | 63.7 | 46.0 | 21.7 | 46.6 | 60.9 | 156 | 63M |
| DAB-DETR-R101 | | 50 | 43.5 | 63.9 | 46.6 | 23.6 | 47.3 | 61.5 | 174 | 63M |
| DAB-DETR-R101* | | 50 | **44.1** | 64.7 | 47.2 | 24.1 | 48.2 | 62.9 | 179 | 63M |
| DETR-DC5-R101 | | 500 | 44.9 | 64.7 | 47.7 | 23.7 | 49.5 | 62.3 | 253 | 60M |
| TSP-RCNN-R101 | ✓ | 96 | 46.5 | 66.0 | 51.2 | 29.9 | 49.7 | 59.2 | 254 | – |
| SMCA-R101 | ✓ | 50 | 44.4 | 65.2 | 48.0 | 24.3 | 48.5 | 61.0 | 218 | 50M |
| Anchor DETR-R101* | | 50 | 45.1 | 65.7 | 48.8 | 25.8 | 49.4 | 61.6 | – | 58M |
| Conditional DETR-DC5-R101 | | 50 | 45.0 | 65.5 | 48.4 | 26.1 | 48.9 | 62.8 | 262 | 63M |
| DAB-DETR-DC5-R101 | | 50 | 45.8 | 65.9 | 49.3 | 27.0 | 49.8 | 63.8 | 282 | 63M |
| DAB-DETR-DC5-R101* | | 50 | **46.6** | 67.0 | 50.2 | 28.1 | 50.5 | 64.1 | 296 | 63M |

Table 2: Results for our DAB-DETR and other detection models. All DETR-like models except DETR use 300 queries, while DETR uses 100. The models with superscript * use 3 pattern embeddings as in Anchor DETR (Wang et al., 2021). We also provide stronger results of our DAB-DETR in Appendix G and Appendix C.

| #Row | Anchor Box (4D) vs. Point (2D) | Anchor Update | $wh$-Modulated Attention | Temperature Tuning | AP |
|---|---|---|---|---|---|
| 1 | 4D | ✓ | ✓ | ✓ | 45.7 |
| 2 | 4D | | ✓ | ✓ | 44.0 |
| 3 | 4D | ✓ | | ✓ | 45.0 |
| 4 | 2D | ✓ | | ✓ | 44.0 |
| 5 | 4D | ✓ | ✓ | | 44.4 |

Table 3: Ablation results for our DAB-DETR. All models are tested over ResNet-50-DC5 backbone and the other parameters are the same as our default settings.

3 and Row 4) and the anchor update introduces 1.7% AP improvement (compare Row 1 and Row 2), which demonstrates the effectiveness of dynamic anchor box design.

After removing modulated attention and temperature tuning, the model performance drops to 45.0% (compare Row 1 and Row 3) and 44.4% (compare Row 1 and Row 5), respectively. Hence fine-grained tuning of positional attentions is of great importance for improving the detection performance as well.

## 6 CONCLUSION

We have presented in this paper a novel query formulation using dynamic anchor boxes for DETR and offered a deeper understanding of the role of queries in DETR. Using anchor boxes as queries leads to several advantages, including a better positional prior with temperature tuning, size-modulated attention to account for objects of different scales, and iterative anchor update for improving anchor estimate gradually. Such a design makes it clear that queries in DETR can be implemented as performing soft ROI pooling layer by layer in a cascade manner. Extensive experiments were conducted and effectively confirmed our analysis and verified our algorithm design.

## ACKNOWLEDGEMENTS

This work was supported by the National Key Research and Development Program of China (2020AAA0104304, 2020AAA0106000, 2020AAA0106302), NSFC Projects (Nos. 61620106010, 62061136001, 61621136008, 62076147, U19B2034, U1811461, U19A2081), Beijing NSF Project (No. JQ19016), Beijing Academy of Artificial Intelligence (BAAI), Tsinghua-Alibaba Joint Research Program, Tsinghua Institute for Guo Qiang, Tsinghua-OPPO Joint Research Center for Future Terminal Technology.

We thank all anonymous reviewers for their valuable comments and suggestions, especially the instructive questions from Reviewer 3.

## ETHICS STATEMENT

Object detection is a fundamental task in computer vision with wide applications. Hence any improvement of this field will yield lots of impacts. To visually perceive and interact with the environment, autonomous vehicles highly depend on this technique and will benefit from any of its improvement. It has also led to advances in medical imaging, word recognition, instance segmentation on natural images, and so on. Therefore a failure in this model could affect many tasks. Our study provides a deeper understanding of the roles of queries in DETR and improves the interpretability of this important submodule in the end-to-end Transformer-based detection framework.

As our model relies on deep neural networks, it can be attacked by adversarial examples. Similarly, as it relies on training data, it may produce biased results induced from training samples. These are common problems in deep learning and our community is working together to improve them. Finally, it is worth noting that detection models, especially face or human detection models, might pose a threat to people's privacy and security if used by someone up to no good.

## REPRODUCIBILITY STATEMENT

We confirm the reproducibility of the results. We have released the source code on Github at https://github.com/IDEA-opensource/DAB-DETR with all materials that are needed to reproduce our results.

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

# Appendix for DAB-DETR

## A TRAINING DETAILS

**Architecture.** Our model is almost the same as DETR which includes a CNN backbone, multiple Transformer (Vaswani et al., 2017) encoders and decoders, and two prediction heads for boxes and labels. We use ImageNet-pretrained ResNet (He et al., 2016) as our backbones, and 6 Transformer encoders and 6 Transformer decoders in our implementations. We follow previous works to report results over four backbones: ResNet-50, ResNet-101, and their 16×-resolution extensions ResNet-50-DC5 and ResNet-101-DC5. As we need to predict boxes and labels in each decoder layer, the MLP networks for box and label predictions share the same parameters across different decoder layers. As inspired by Anchor DETR, we also leverage multiple pattern embeddings to perform multiple predictions at one position and the number of patterns is set as 3 which is the same as Anchor DETR. We also leverage PReLU (He et al., 2015) as our activations.

Following Deformable DETR and Conditional DETR, we use 300 anchors as queries. We select 300 predicted boxes and labels with the largest classification logits for evaluation as well. We also use focal loss (Lin et al., 2020) with $\alpha = 0.25, \gamma = 2$ for classification. The same loss terms are used in bipartite matching and final loss calculating, but with different coefficients. Classification loss with coefficient 2.0 is used in bipartite matching but 1.0 in the final loss. L1 loss with coefficient 5.0 and GIOU loss (Rezatofighi et al., 2019) with coefficient 2.0 are consistent in both the matching and the final loss calculation procedures. All models are trained on 16 GPUs with 1 image per GPU and AdamW (Loshchilov & Hutter, 2018) is used for training with weight decay $10^{-4}$. The learning rates for backbone and other modules are set to $10^{-5}$ and $10^{-4}$, respectively. We train our models for 50 epochs and drop the learning rate by 0.1 after 40 epochs. All models are trained on Nvidia A100 GPU. We search hyperparameters with batch size 64 and all results in our paper are reported with batch size 16. For better reproducing our results, we provide the memory needed and batch size/GPU in Table 4.

**Dataset.** We conduct the experiments on the COCO (Lin et al., 2014) object detection dataset. All models are trained on the `train2017` split and evaluated on the `val2017` split.

| Model | Batch Size/GPU | GPU Memory (MB) |
|---|---|---|
| DAB-DETR-R50 | 2 | 6527 |
| DAB-DETR-R50* | 1 | 3573 |
| DAB-DETR-R50-DC5 | 1 | 13745 |
| DAB-DETR-R50-DC5* | 1 | 15475 |
| DAB-DETR-R101 | 2 | 6913 |
| DAB-DETR-R101* | 1 | 4369 |
| DAB-DETR-R101-DC5 | 1 | 13148 |
| DAB-DETR-R101-DC5* | 1 | 16744 |

Table 4: GPU memory usage of each model.

## B COMPARISON OF DETR-LIKE MODELS

In this section, we provide a more detailed comparison of DETR-like models, including DETR (Carion et al., 2020), Conditional DETR (Meng et al., 2021), Anchor DETR (Wang et al., 2021), Deformable DETR (Zhu et al., 2021), our proposed DAB-DETR, and DAB-Deformable-DETR. Their model designs are illustrated in Fig. 8. We will discuss the difference between previous models and our models.

Anchor DETR (Wang et al., 2021) improves DETR by introducing 2D anchor points, which are updated layer by layer. It shares a similar motivation with our work. But it leaves the object scale information unconsidered and thus cannot modulate the cross-attention to make it adapt to objects of different scales. Moreover, the positional queries in its framework are of high dimension and passed to the self-attention modules in all layers without any adaptation. See the brown-colored part in Fig. 8 (d) for details. This design might be sub-optimal as the self-attention modules cannot leverage the refined anchor points in different layers.

Deformable DETR (Zhu et al., 2021) introduces 4D anchor boxes and updates them layer by layer, which is called *iterative bounding box refinement* in its paper. Its algorithm is mainly developed based on deformable attention, which requires reference points to sample attention points and meanwhile utilizes box width and height to modulate attention areas. However, as *iterative bounding box refinement* is closely coupled with the special design of deformable attention, it is nontrivial to apply it to general Transformer decoder-based DETR models. This is probably the reason why few works after Deformable DETR adopt this idea. Moreover, the position queries in Deformable DETR are passed to both the self-attention modules and the cross-attention modules in all layers without any adaptation. See the brown-colored part in Fig. 8 (e) for details. As a result, both its self-attention modules and cross-attention modules cannot fully leverage the refined anchor boxes in different layers.

To verify our analysis, we develop a variant of Deformable-DETR by formulating its queries as dynamic anchor boxes as in DAB-DETR. We call this variant as DAB-Deformable-DETR, which is illustrated in Fig. 8 (f). Under exactly the same setting using R50 as the backbone, DAB-Deformable-DETR improves Deformable-DETR by 0.5 AP (46.3 to 46.8) on COCO. See Table 5 for the performance comparison and Sec. C for more implementation details.

Dynamic DETR (Dai et al., 2021) is another interesting improvement of DETR. It also leverages anchor boxes to pool features, but it uses ROI pooling for feature extraction, which makes it less general to DETR-like models compared with our dynamic anchor boxes. Moreover, compared with cross-attention in Transformer decoders, which performs global feature pooling in a soft manner (based on attention maps), the ROI pooling operation only performs local feature pooling within a ROI window. In our opinion, the ROI pooling operation can help faster convergence as it enforces each query to associate with a specific spatial position. But it may lead to a sub-optimal result due to its ignorance of the global context outside a ROI window.

## C   DAB-DEFORMABLE-DETR

To further demonstrate the effectiveness of our dynamic anchor boxes, we develop DAB-Deformable-DETR by adding our dynamic anchor boxes design to Deformable DETR (Zhu et al., 2021) [2]. The difference between Deformable DETR and DAB-Deformable-DETR is shown in Fig. 8 (e) and (f). The results of Deformable DETR and DAB-Deformable-DETR are shown in Table 5. With no more than 10 lines of code modified, our DAB-Deformable-DETR (row 4) results in a significant performance improvement (+0.5 AP) compared with the original Deformable DETR (row 3). All other settings except the query formulation are exactly the same in this experiment.

We also compare the speed of convergence in Fig. 9. It shows that our proposed dynamic anchor boxes speed up the training as well (left in Fig. 9). We believe one of the reasons for better performance is the update of learned queries. We plot the change of total loss, which is the sum-up of losses of all decoder layers, during training in the middle figure of Fig. 9. Interestingly, it shows that the total loss of DAB-Deformable-DETR is larger than Deformable DETR. However, the loss of the final layer of DAB-Deformable-DETR is lower than that in Deformable DETR (right in Fig. 9), which is a good indicator of the better performance of DAB-Deformable-DETR as the inference result only takes from the last layer.

## D   ANCHORS VISUALIZATION

We visualize the learned anchor boxes in Fig. 10. When learning anchor points as queries, the learned points are distributed evenly around the image, while the centers seem to distribute randomly when learning anchor boxes directly. This might be because the centers are coupled with anchor sizes. The right-most figure shows the visualization of the learned anchor boxes. We only show a partial set for visualization clarity. Most boxes are of medium size and no particular pattern is found in the distribution of boxes.

---

[2]We used the open-source implementation from https://github.com/fundamentalvision/Deformable-DETR

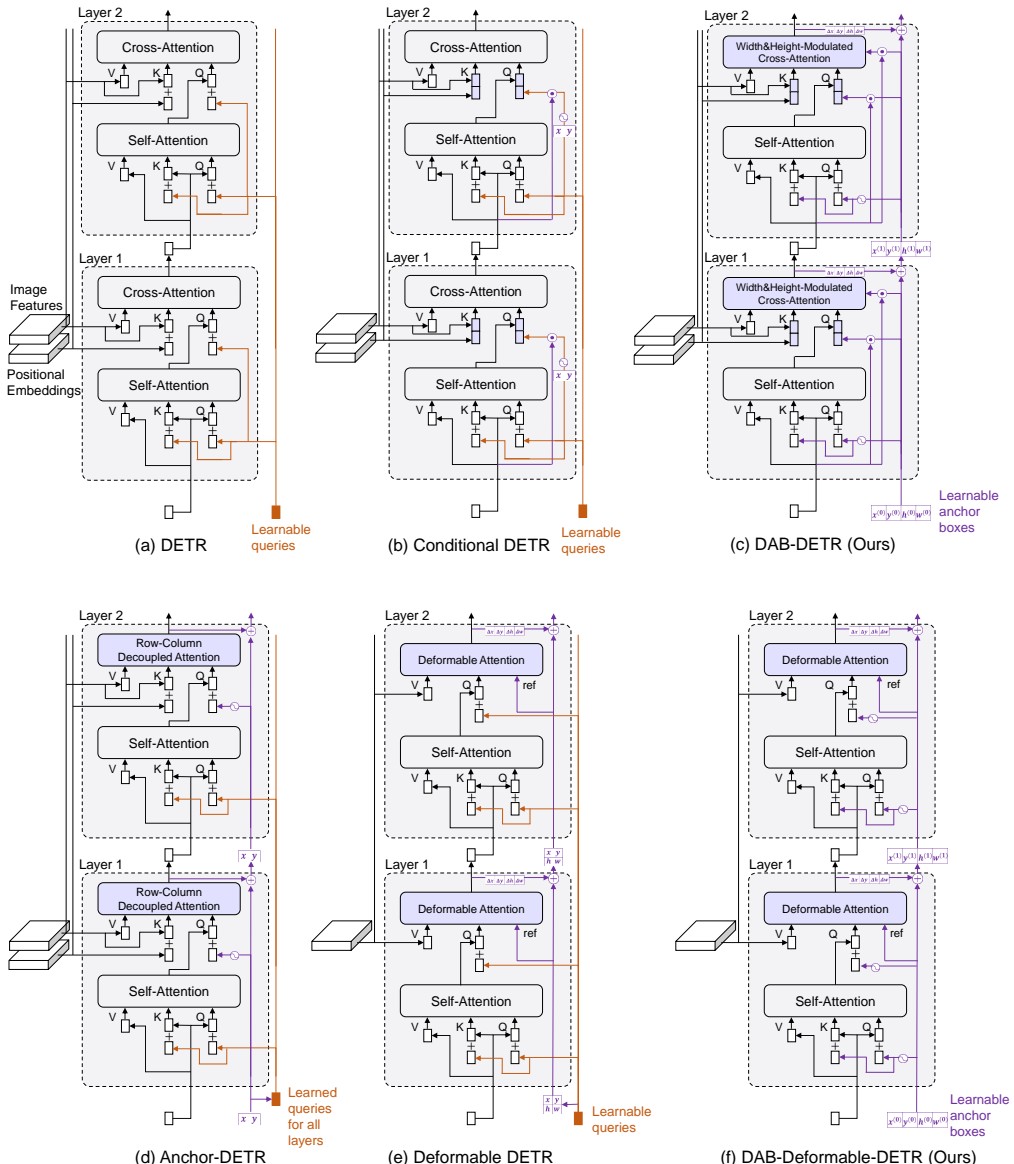

Figure 8: Comparison of DETR-like models. For clarity, we only show two layers of Transformer decoder and omit the FFN blocks. We mark the modules with difference in purple and marked the learned high-dimensional queries in brown. DAB-DETR (c) is proposed in our paper, and DAB-Deformable-DETR (f) is a variant of Deformable DETR modified by introducing our dynamic anchors boxes. All previous models (a,b,d,e) leverage high-dimensional queries (shaded in brown) to pass positional information to each layers, which are semantic ambiguous and are not updated layer by layer. In contrast, DAB-DETR (c) directly uses dynamically updated anchor boxes to provide both a reference query point $(x, y)$ and a reference anchor size $(w, h)$ to improve the cross-attention computation. DAB-Deformable-DETR (f) uses dynamically updated anchor boxes to formulate its queries as well.

| # row | Model | AP | AP$_{50}$ | AP$_{75}$ | AP$_S$ | AP$_M$ | AP$_L$ | Params |
|---|---|---|---|---|---|---|---|---|
| 1 | Deformable DETR | 43.8 | 62.6 | 47.7 | 26.4 | 47.1 | 58.0 | 40M |
| 2 | Deformable DETR+ | 45.4 | 64.7 | 49.0 | 26.8 | 48.3 | 61.7 | 40M |
| 3 | Deformable DETR+ (open source) | 46.3 | 65.3 | 50.2 | 28.6 | 49.3 | 62.1 | 47M |
| 4 | DAB-Deformable-DETR(Ours) | **46.8** | **66.0** | **50.4** | **29.1** | **49.8** | **62.3** | 47M |

Table 5: Comparison of the results of Deformable DETR and DAB-Deformable-DETR. The models in row 1 and row 2 are copied from the original paper, and the models in row 3 and row 4 are tested under the same standard R50 multi-scale setting. Deformable DETR+ means the Deformable DETR model with iterative bounding box refinement and the result of Deformable DETR+ (open source) is reported by us using the open-source code. The only difference between row 3 and row 4 is the formulation of queries.

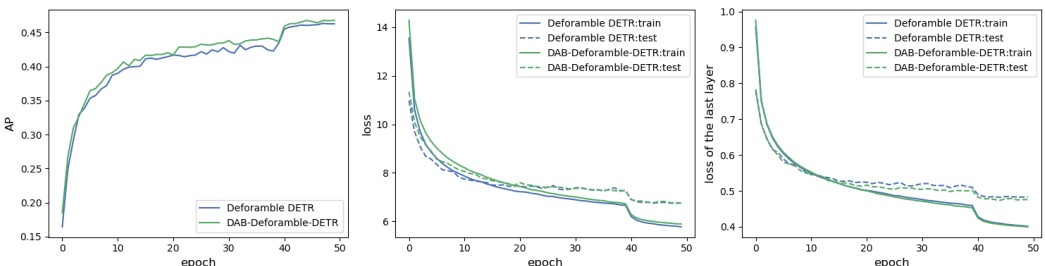

Figure 9: Comparison of the training of Deformable DETR and DAB-Deformable-DETR models. We plot the change of AP (left), the loss of all layers (middle), and the loss of the last layer (right) during training, respectively. With no more than 10 lines of code modified, DAB-Deformable-DETR results in a better performance compared with the original Deformable DETR model (see the left figure). While the loss of all layers of DAB-Deformable-DETR is larger than that in Deformable DETR (see the middle figure), our models have a lower loss of the last layer (see the right figure), which is the most important as the inference result only takes from the last layer. The two models are tested under the same standard R50 multi-scale setting.

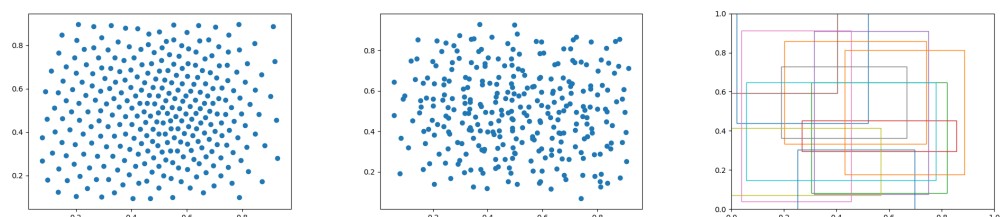

Figure 10: Learned anchor points when learning 2D coordinates only (left), and anchor center points (middle) and partial anchor boxes (right) when learning anchor boxes directly.

# E    RESULTS WITH DIFFERENT TEMPERATURES

Table 6 shows the results of models using different temperatures in the positional encoding function. As larger temperature generates more flattened attention maps, it leads to better performances for larger objects. For example, the model with $T = 2$ and the model with $T = 10000$ have similar AP results, but the former has better performances on AP$_S$ and AP$_M$, while the latter works better on AP$_L$, which also validates the role of positional priors in DETR.

| Temperature | AP | $AP_{50}$ | $AP_{75}$ | $AP_S$ | $AP_M$ | $AP_L$ |
|---|---|---|---|---|---|---|
| 2 | 39.6 | 60.7 | 41.9 | 19.3 | 43.3 | 58.0 |
| 5 | 40.0 | 61.1 | 42.1 | 19.5 | 43.4 | 58.9 |
| 10 | 40.0 | 61.1 | 42.3 | 19.7 | 43.5 | 59.3 |
| 20 | 40.1 | 61.1 | 42.8 | 19.8 | 43.7 | 58.6 |
| 50 | 39.8 | 61.0 | 42.2 | 19.7 | 43.2 | 58.8 |
| 100 | 39.8 | 60.8 | 42.1 | 19.3 | 43.3 | 58.4 |
| 10000 | 39.5 | 60.7 | 41.7 | 18.9 | 42.6 | 58.9 |

Table 6: Comparison of models with different temperatures. All models are trained with the ResNet-50 backbone, batch size $64$, no multiple pattern embeddings, and no modulated attentions. Default Settings are used for the rest of the parameters.

## F  RESULTS WITH LESS DECODER LAYERS

Table 7 shows the results of models with different decoder layers. All models are trained under our standard ResNet-50-DC setting except the number of decoder layers.

| decoder layers | GFLOPs | Parmas | AP | $AP_{50}$ | $AP_{75}$ | $AP_S$ | $AP_M$ | $AP_L$ |
|---|---|---|---|---|---|---|---|---|
| 2 | 202 | 36M | 40.2 | 59.0 | 42.9 | 22.2 | 43.5 | 55.4 |
| 3 | 206 | 38M | 43.9 | 63.4 | 47.4 | 24.6 | 47.8 | 60.5 |
| 4 | 210 | 40M | 44.9 | 64.5 | 48.2 | 25.9 | 48.5 | 61.0 |
| 5 | 213 | 42M | 45.2 | 65.5 | 48.6 | 26.6 | 48.9 | 62.3 |
| 6 | 216 | 44M | 45.7 | 66.2 | 49.0 | 26.1 | 49.4 | 63.1 |

Table 7: Comparison of models with different number of decoder layers. All models are trained under our standard ResNet-50-DC setting except the number of decoder layers.

## G  FIXED $x, y$ FOR BETTER PERFORMANCE

We provide in this section an interesting experiment. As we all know, all box coordinates $x, y, h, w$ are learned from data. When we fix $x, y$ of the anchor boxes with the random initialization, the model's performance increases consistently. The comparison of standard DAB-DETR and DAB-DETR with fixed $x, y$ coordinates is shown in Table 8. Note that we only fix $x, y$ at the first layer to prevent them from learning information from data. But $x, y$ will be updated in other layers. We conjecture that the randomly initialized and fixed $x, y$ coordinates can help to avoid overfitting, which may account for this phenomenon.

## H  COMPARISON OF BOX UPDATE

To further demonstrate the effectiveness of our dynamic anchor box design, we plot the layer-by-layer update result of boxes of DAB-DETR and Conditional DETR in Fig. 11. All DETR-like models have a stacked layers structure. Hence the outputs of each layer can be viewed as a refining procedure. However, due to the high-dimensional queries that are shared across all layers, the update of queries between layers is not stable. As shaded in yellow in Fig. 11 (b), some boxes predicted in the latter layers are worse than their previous layers.

## I  ANALYSIS OF FAILURE CASES

Fig. 12 presents some samples where our model does not predict well. We find our model may have some troubles when facing dense objects, very small objects, or very large objects in an image. To

| Model | AP | $AP_{50}$ | $AP_{75}$ | $AP_S$ | $AP_M$ | $AP_L$ |
|---|---|---|---|---|---|---|
| DAB-DETR-R50* | 42.6 | 63.2 | 45.6 | 21.8 | 46.2 | 61.1 |
| DAB-DETR-R50*-fixed$x$&$y$ | **42.9**(+0.3) | 63.7 | 45.3 | 22.0 | 46.8 | 60.9 |
| DAB-DETR-DC5-R50 | 44.5 | 65.1 | 47.7 | 25.3 | 48.2 | 62.3 |
| DAB-DETR-DC5-R50-fixed$x$&$y$ | **44.7**(+0.2) | 65.3 | 47.9 | 24.9 | 48.2 | 62.0 |
| DAB-DETR-DC5-R50* | 45.7 | 66.2 | 49.0 | 26.1 | 49.4 | 63.1 |
| DAB-DETR-DC5-R50*-fixed$x$&$y$ | **45.8**(+0.1) | 66.5 | 48.9 | 26.4 | 49.6 | 62.7 |
| DAB-DETR-R101* | 44.1 | 64.7 | 47.2 | 24.1 | 48.2 | 62.9 |
| DAB-DETR-R101*-fixed$x$&$y$ | **44.8**(+0.7) | 65.4 | 48.2 | 25.1 | 48.9 | 63.1 |
| DAB-DETR-DC5-R101* | 46.6 | 67.0 | 50.2 | 28.1 | 50.5 | 64.1 |
| DAB-DETR-DC5-R101*-fixed$x$&$y$ | **46.7**(+0.1) | 67.3 | 50.7 | 27.3 | 50.9 | 64.1 |

Table 8: Comparison of DAB-DETR and DAB-DETR with fixed anchor centers $x, y$. When fixing $x, y$ of queries with random values, the performance of the models is improved consistently. The models with superscript * use 3 pattern embeddings as in Anchor DETR.

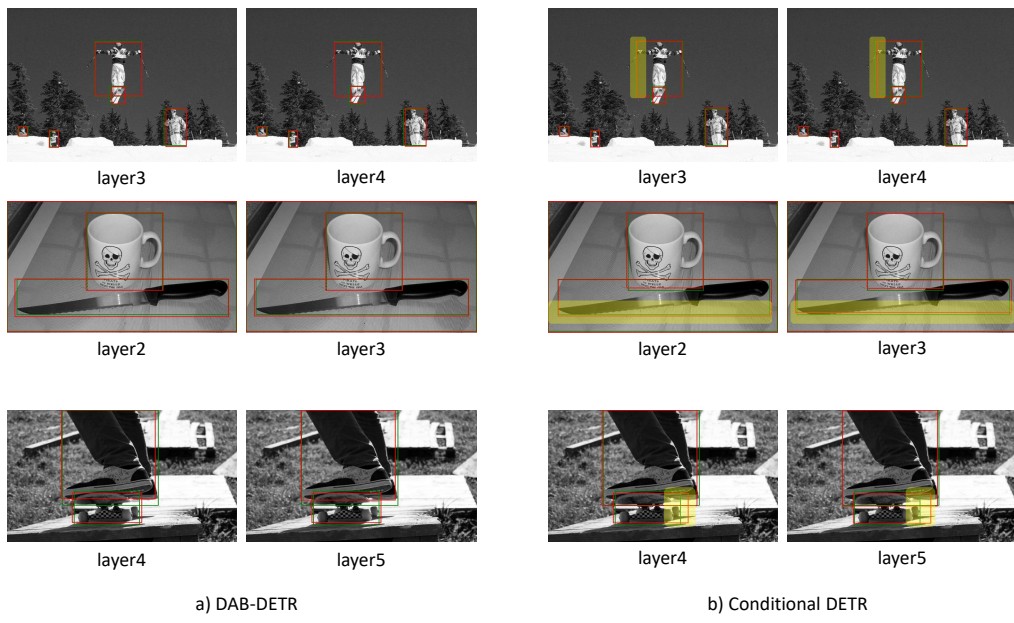

Figure 11: We compare the layer-by-layer update of boxes of DAB-DETR (a) and Conditional DETR (b). The green boxes are ground truth annotations while the red boxes are model predictions. The boxes of Conditional DETR have larger variances and we mark some boundaries of boxes with a large change in yellow.

improve the performance of our model, we will introduce a multi-scale technique into our model to improve the detection performance on small and large objects.

## J COMPARISON OF RUNTIME

We compare the runtime of DETR, Conditional DETR, and our proposed DAB-DETR in Table 9. Their runtime speeds are reported on a single Nvidia A100 GPU. Our DAB-DETR has a similar inference speed but better performance compared with Conditional DETR, which is our direct competitor.

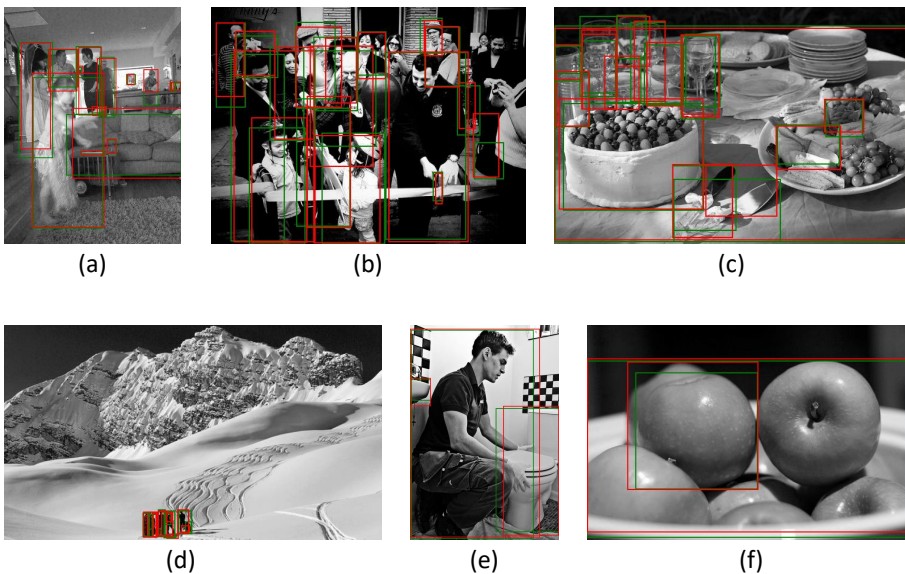

Figure 12: We visualize some images where our model does not predict well, including dense objects (a,b,c), very small objects (d), and very large objects (e,f). The green boxes are ground truth annotations while red boxes are predictions of models.

| Model | time(s/img) | epoches | AP | AP$_{50}$ | AP$_{75}$ | AP$_S$ | AP$_M$ | AP$_L$ | Parmas |
|---|---|---|---|---|---|---|---|---|---|
| DETR-R50 | 0.048 | 500 | 42.0 | 62.4 | 44.2 | 20.5 | 45.8 | 61.1 | 41M |
| Conditional DETR-R50 | 0.057 | 50 | 40.9 | 61.8 | 43.3 | 20.8 | 44.6 | 59.2 | 44M |
| DAB-DETR-R50 | 0.059 | 50 | 42.2 | 63.1 | 44.7 | 21.5 | 45.7 | 60.3 | 44M |
| DETR-R101 | 0.074 | 500 | 43.5 | 63.8 | 46.4 | 21.9 | 48.0 | 61.8 | 60M |
| Conditional DETR-R101 | 0.082 | 50 | 42.8 | 63.7 | 46.0 | 21.7 | 46.6 | 60.9 | 63M |
| DAB-DETR-R101 | 0.085 | 50 | 43.5 | 63.9 | 46.6 | 23.6 | 47.3 | 61.5 | 63M |

Table 9: Comparison of the runtime of DETR, Conditional DETR, and our proposed DAB-DETR. All speeds are reported on a single Nvidia A100 GPU.

## K    COMPARISON OF MODEL CONVERGENCE

We present convergence curves of DETR, Conditional DETR, and our DAB-DETR in Fig. 13. All models are trained under the standard R50 (DC5) setting. The results demonstrate the effectiveness of our model. Our DAB-DETR is trained with our $fix\ x\&y$ variants. see Appendix G for more details about the $fix\ x\&y$ results. Both Conditional DETR and DAB-DETR use 300 queries, while DETR leverages 100 queries.

Our DAB-DETR converges faster than Conditional DETR, especially in early epochs, as shown in Fig. 13.

## L    VISUALIZATION RESULTS OF ITERATIVE BOX UPDATE

We present more visualization results of iterative box update in Fig. 14 and Fig. 15. The initial anchors, anchors updated after the first decoder layer, and the anchors predicted from the last decoder layer are plotted in the first, the second, and the third columns, respectively.

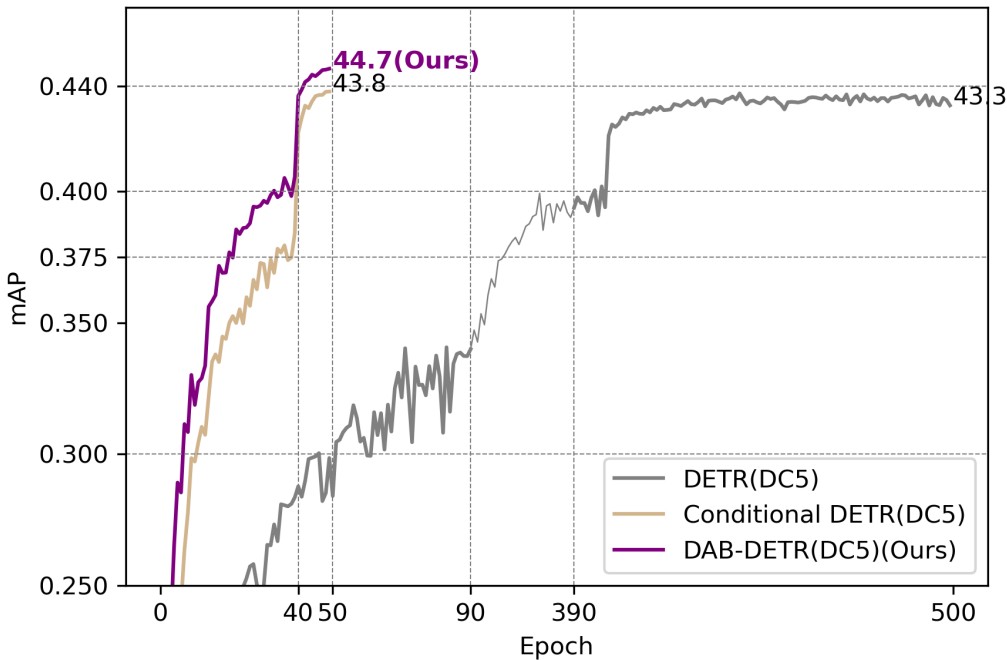

Figure 13: Convergence curves of DETR, Conditional DETR, and our DAB-DETR. All models are trained under the R50 (DC5) setting.

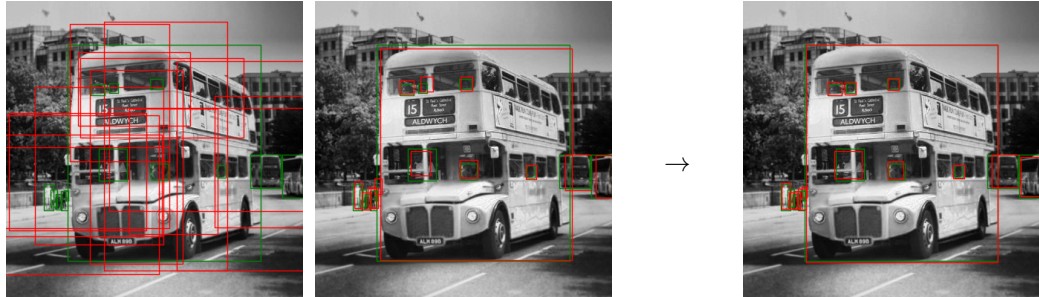

Figure 14: Visualizations for layer-by-layer anchor box update. We plot the initial anchor boxes (left), anchor boxes after the first decoder layer (middle), and the output of the last decoder layer (right), respectively. The green boxes are ground truth annotations, while the red boxes are predictions of our model. The results are obtained using the ResNet-50 backbone. More visualizations are available in Fig. 15.

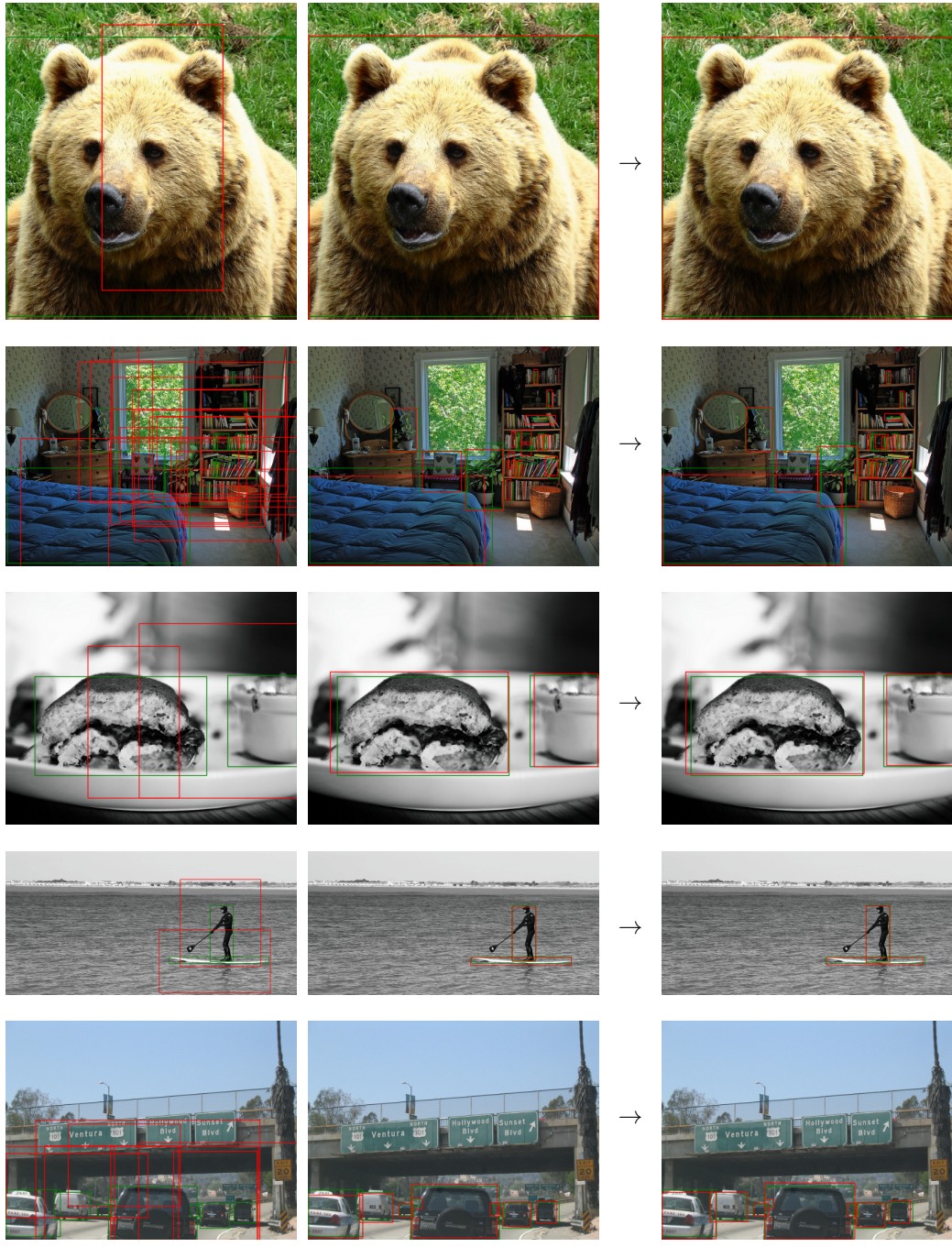

Figure 15: More visualizations for layer-by-layer anchor box update. We plot the initial anchor boxes (left), anchor boxes after the first decoder layer (middle), and the output of the last decoder layer (right), respectively. The green boxes are ground truth annotations, while the red boxes are predictions of our model. The results are obtained using the ResNet-50 backbone.

