# OpenReview forum: "DAB-DETR: Dynamic Anchor Boxes are Better Queries for DETR"
_ICLR.cc/2022/Conference — ICLR 2022 Poster_

### Official Review · Reviewer_zkuH · 2021-11-02

**Correctness:** 3
**Technical Novelty And Significance:** 2
**Empirical Novelty And Significance:** 2
**Recommendation:** 5
**Confidence:** 5

**Main Review:**

### Pros

- This paper provides a systematic comparison between several detection transformer variants, from the perspective of positional encoding and attention mechanism, as summarized in Table 1.
- The experiment results are promising, and DAB-DETR achieves state-of-the-art performance on COCO 2017 dataset.

### Cons

- The novelty of this paper is very limited, the idea of using explicit positional priors to accelerate training and improve explainability has already been adopted in Deformable DETR, Conditional DETR, and Anchor DETR.
- This is incremental work. Almost every part of the method can be found in existing work. For example, the iterative update of 4-D anchor boxes resembles the update of reference points in Deformable DETR, the decoupling of the content and positional queries resembles Conditional DETR, and the explicit anchor design resembles Anchor DETR. This paper only makes a combination of those methods, while offering little novel insights.
- The observations in Section 3 are almost obvious, and the reviewer did not feel that new information was brought to the table after reading.
- The conclusion in Section 3 is somewhat weak, as the authors described "the multiple mode property of queries in DETR **is likely** the root cause for its slow training". And the method is proposed based on this somewhat invalid conclusion, which makes the motivation less clear, and the method less convincing. Moreover, the observations in Figure 4 are not supported by quantitative experiments, and it is difficult to give conclusions from the qualitative visualization of 4 out of hundreds of anchors.

**Summary Of The Paper:**

This paper proposes DAB-DETR, a DETR variant that incorporates dynamic anchor boxes for object queries. Based on the analysis of the learned queries in DETR, the authors conclude the slow convergence of DETR is likely caused by the multiple mode property of the queries. To solve the problem, explicit positional priors (box coordinates) are used to improve query-to-feature similarity and accelerate model convergence. Experiments and analysis show that DAB-DETR achieves promising results on the object detection task.

**Summary Of The Review:**

This paper proposes a DETR variant that incorporates dynamic anchor boxes to accelerate model convergence. Though promising result has been achieved, the method lacks novelty and the motivation is not very clear. Thus, the paper is not ready for publication.

---

> ### Author Response · Authors · 2021-11-16
> **Responses to Reviewer zkuH (R3) part 1**
>
> Thanks for your careful review and suggestions. We're sorry that we didn't articulate our contributions clearly in the main text, hence we reclarify our novelty both below and in the revised version. We will first summarize what we have done this week to address your key concerns and then provide detailed responses for each weakness you have pointed out.
>
> ### Summary of your key concerns and our improvements:
> 1. You think the novelty of explicit positional priors is limited.
> - Our response: We added a detailed comparison of our DAB-DETR and other DETR-like models in Appendix B and reclarify our novelty below.
> 2. You think this is incremental work.
> - Our response: We will show the novelty of our dynamic anchor boxes by 1) comparing with all previous works in Appendix B, 2) designing a DAB-Deformable-DETR with better performance in Appendix C, and 3) reclarifying the novelty of our width-height modulated attention and temperature tuning.
> 3. You think the conclusion of Sec. 3 is weak.
> - Our response: We further verified our assumption by adding a new experiment in Sec. 3, where we replace the query formulation with dynamic anchor boxes only and result in better performance.
>
> ### Detailed responses to your concerns item-by-item.
>
> **1.**
> - We agree the use of explicit positional priors has been adopted in previous works. However, the formulation of queries with dynamic anchor boxes is what we highlight and novel.
> - Moreover, the width-height modulated attention and temperature tuning in our paper are still novel.
> **2.**
> - First, we want to point out the width-height modulated attention and temperature tuning are still novel for making cross-attention adaptive to objects of different scales, which was overlooked in previous studies.
> - We added a detailed comparison of several DETR-like models in Appendix B and Fig. 8. We compared DETR, Conditional DETR, Deformable DETR, Anchor DETR, our DAB-DETR, and an extra DAB-Deformable-DETR, which is made by replacing the query formulation of Deformable DETR with our dynamic anchor boxes (The deformable DETR didn't formulate queries as anchor boxes.).
> - We agree the iterative update of 4D boxes was leveraged in the Deformable DETR, but the use of boxes is closely coupled with the special deformable attention mechanism, which makes it nontrivial to use 4D boxes in other DETR-like models. As a result, few work after Deformable DETR adopts its 4D box update approach until our DAB-DETR. Moreover, Deformable DETR uses shared positional queries in both the self-attention and cross-attention modules in all layers, which is not a proper algorithm design in our opinion and can lead to sub-optimal results. By contrast, our dynamic anchor box formulation is more general and can be used in all DETR-like models.
> - To verify our analysis, we developed a simple variant of Deformable DETR by replacing its query formulation with our dynamic anchor boxes and achieved both faster convergence and better performance, which is shown in Appendix C. We name this new model DAB-Deformable-DETR. The comparison of the results of Deformable DETR and DAB-Deformable-DETR is presented both in Table 5 and below. All other settings except the query formulation are exactly the same in this experiment.
>
> | Model               | AP   | AP$_{50}$ | AP$_{75}$ | AP$_{S}$  | AP$_{M}$  | AP$_{L}$  |
> | ------------ | ---- | ---- | ---- | ---- | ---- | ---- |
> | Deformable DETR     | 46.3 | 65.3 | 50.2 | 28.6 | 49.3 | 62.1 |
> | DAB-Deformable-DETR | 46.8 | 66.0 | 50.4 | 29.1 | 49.8 | 62.3 |
>
> Table R3.1: Comparison of the results of Deformable DETR and DAB-Deformable-DETR.
>
> - The Deformable DETR learns high-dimensional queries as well, which will not and cannot be updated between layers, as shaded in brown in Fig. 8 (e). (If you agree with our illustration above, you can skip the details in this parentheses. Or, you can see the `query_embeds` variable in https://github.com/fundamentalvision/Deformable-DETR/blob/main/models/deformable_detr.py#156, the `query_embed` variable in https://github.com/fundamentalvision/Deformable-DETR/blob/main/models/deformable_transformer.py#181, and the `query_pos` variable in https://github.com/fundamentalvision/Deformable-DETR/blob/main/models/deformable_transformer.py #326, #338, # 295, #297, and #303. The `query_pos`, which means the high-dimensional learnable queries, are shared across all layers without an update.)  The position queries are passed to both the self-attention modules and the cross-attention modules in all layers without any adaptation, as shown in Fig. 8 (e). As the result, both its self-attention modules and cross-attention modules cannot fully leverage the refined anchor boxes in different layers.
>
>
> To be continued...

---

> ### Author Response · Authors · 2021-11-16
> **Responses to Reviewer zkuH (R3) part 2**
>
> Continued to responses part 1.
>
> ### Detailed responses to your concerns item-by-item. (continued)
> **2. (continued)**
> - We agree both two algorithms (Deformable DETR and DAB-DETR) share a similar spirit of performing iterative 4D box update. Deformable DETR does the update on reference anchors only, whereas our DAB-DETR does it on the queries of the whole decoder (see Fig. 8 (e) and (f)). The former design is closely coupled with the special deformable attention mechanism and cannot be easily applied to other DETR-like models. In contrast, the design of DAB-DETR is more general and can be used to improve most DETR-like models.
> - Now we turn to Anchor-DETR. It's worth noting that Anchor DETR was published on arXiv on Sep. 15th, which is only 3 weeks before our submission. Hence it's a concurrent work with us. It's for a fair comparison that we adopted its design of multiple patterns for one query in our submission, which is an orthogonal idea to our proposed anchor box formulation.
> - Even so, they still have high-dimensional queries for all layers without an update. Moreover, they only formulate queries as 2D anchor points, which can result in inferior performance as shown in our main table and ablations.
> - In summary, we believe our paper has enough novelty with a neat and more general solution. We hope this discussion can better clarify our novelty.
>
> **3&4.**
> - As cons 3 and 4 are both about Sec. 3, we respond to them together.
> - Thanks for pointing out that our conclusion in Sec. 3 is weak. We improved it by adding a new experiment. To verify the impact of multi-mode of old queries, we replace the original DETR queries with the dynamic anchor box formulation, without other techniques like 300 queries, focal loss, or special attention. We name the new model DETR+DAB, which is different from our DAB-DETR as it doesn't have the width-height modulated attention and the temperature tuning. All settings are the same for the two models. The results in Fig.3 b) demonstrate the effectiveness of our dynamic anchor box formulation. It also provides convincing evidence that the multi-mode property of learned queries is at least one of the key problems of slow convergence of DETR.
>
> ### Summary:
> Thanks for your valuable questions and suggestions. Your questions are more about our novelty and motivation. Hope our response can convince you of our novelty. Moreover, we provided new experiments in Sec.3, which we believe can verify our conclusions.

---

> > ### Comment · Reviewer_zkuH · 2021-11-24
> > **Some concerns should be further addressed**
> >
> > I appreciate the authors' efforts, but I still have the following concerns.
> >
> > * The performance claims in both abstract and introduction, i.e., "AP 45.7% using R50 as backbone trained in 50 epochs" and "The proposed method can achieve 45.7% AP when using a single ResNet-50 (He et al., 2016) model as backbone for training 50 epochs.", are exaggerations to me. The authors should at least mention the use of Dilated Convolution (DC5), not to say the use of 3 pattern embeddings as in Anchor DETR, to obtain 45.7% on AP.
> > * Questions about Table 1
> >   * It's better to include SMCA in this table, since it also uses 4D explicit positional prior and Size-Modulated Attention.
> >   * I think the "Update Learned Queries?" is kind of misleading, since basically all DETR-like object detectors update the content queries layer-by-layer. I would suggest the authors specify that the queries here indicate query positional embedding, for example, "Update Learned Positional Queries?"  or "Update Learned Query Positions?"
> > * Is the explicit query positional embedding formulation better than the query positional embedding formulation with implicit location information, or does the performance gain only come from the dynamic update of the query positional embedding?
> >   * From Figure 8 (e) and (f), it can be seen that the only difference between DAB-Deformable-DETR and Deformable DETR is how the positional information is encoded. Specifically, (1) the initial reference box in DAB-DETR and DAB-Deformable-DETR is directly learned (as did in Sparse RCNN), while the initial reference box in Deformable DETR is learned by projecting the learnable positional queries (let's denote the projection function as `proj`). (2) The learned reference boxes in  DAB-DETR and DAB-Deformable-DETR are encoded by **predefined sine function** before being added to the content queries, while in Deformable DETR, the learnable positional embedding added to the content queries can be seen as the learned reference boxes encoded by the **inverse function of the learned**  `proj` **function**. (3) the positional embedding in DAB-DETR is updated layer by layer, while the positional embedding in Deformable DETR is fixed.
> >   * The first difference, I believe, is minor. While the contribution of the other two differences is not clear. Can the authors provide the experiment results of (1)  DAB-DETR and DAB-Deformable-DETR with learned projection function (in replace of the predefined sine function) for encoding the reference boxes? (2) Deformable-DETR with a layer-wise update on the positional embedding?
> >   * The parameter numbers of Deformable DETR+ (open source) and DAB-Deformable-DETR(Ours) in Table 5 seem to be wrong.
> > * About the novelty of this paper
> >   * From Table 1 and Figure 8, it can be seen that the only thing not used by existing detection transformers is the "Update Learned Positional  Query?". But still, it is more like a trick than a novel method.
> >   * Width & height modulated Gaussian kernel is also adopted in SMCA and the temperature turning is more like a trick.
> >   * "but the use of boxes is closely coupled with the special deformable attention mechanism, which makes it nontrivial to use 4D boxes in other DETR-like models" I could not agree with this. The residual of the 4D boxes in Deformable DETR are generated by applying the regression head (which all detection transformers have) on the content queries. Thus the iterative update of 4D boxes in Deformable DETR does not rely on the Deformable Attention mechanism, and not even on the attention mechanism. Actually, Sparse RCNN, which does not use a DETR-like architecture, also adopts a similar iterative update of 4D boxes.
> >
> > * "As the decoder embeddings are updated by **linearly combining** the image features, they are in the same space as the image features. Hence the root cause is likely due to the learnable queries." This line is confusing to me, and FFN in the decoder also contains non-linear activations.

---

> > > ### Author Response · Authors · 2021-11-27
> > > **Responses to your further concerns (1/4)**
> > >
> > > Thanks for your encouraging response. We are glad to see you agree with some of our explanations and appreciate your more detailed questions and suggestions which will help us further improve the article. As before, we first summarize our responses to your key concerns and then provide detailed responses for each concern you list.
> > >
> > > ### Summary of your key concerns and our responses:
> > > 1. You provide some writing suggestions. (your concerns 1, 2, and 5.)
> > >
> > > Our response: We appreciate your valuable comments and will modify them accordingly in the next version.
> > >
> > > 2. You present a detailed analysis of our DAB-Deformable-DETR and wonder where the performance gain of DAB-Deformable-DETR comes from. (your concern 3.)
> > >
> > > Our response: We point out the flaw in your analysis and provide new experimental results (as you suggested) to verify the strength of our dynamic anchor box formulation.
> > >
> > > 3. You have some new concerns about the novelty of our paper. (your concerns 4.1-4.3.)
> > >
> > > Our response:
> > >
> > > - We point out the inaccuracy of your statement in your concern 4.1 and reclarify the novelty of our dynamic anchor box formulation.
> > > - We present the difference between our Width&Height-Modulated Attention and the Modulated Gaussian Kernel in SMCA for your concern 4.2. Then we reclarify the novelty of our Width&Heigh-Modulated Attention.
> > > - We point out the deviations in your understanding of the sentence you cited and explain why our dynamic anchor box is a nontrivial formulation for DETR-like models. (for your concern 4.3.)
> > >
> > > ### Responses to your concerns item-by-item.
> > > 1&2. Writing suggestions.\
> > > Both two comments are about writing, hence we respond to them together. We appreciate your suggestions and we will modify them accordingly in the next version. More specifically, we will mention DC5 when claiming the performance, add SMCA to our Table 1, and changing “Update Learned Queries?” to “Update Learned Positional Queries?”. Your suggestions are of great help for us to make this article clearer and more precise.
> > >
> > > 3 . Ablations for positional query embedding. \
> > > A. We assume you agreed with the difference between our dynamic anchor box and other methods, and recognized the performance improvement and clear semantics of this formulation.  \
> > > B. You are trying to conclude the reason for DAB-Deformable-DETR’s performance gain to a specific module like sine function or iterative update by analyzing the detailed difference between our DAB-Deformable-DETR and Deformable DETR. **We hope you will not conclude our contributions to one point or one module based on the analysis and comparison**, because the comparison is not representative and the analysis is incomplete. \
> > > C. Why do we say the comparison you suggested is not representative?
> > > - If your intention is not to conclude our contribution to the sine function or the query update, you can skip the whole C paragraph.
> > > - The use of boxes of Deformable DETR is rooted in the special design of deformable attention. For the original DETR, what we need are high-dimensional vectors for queries. It is nontrivial to introduce the dynamic anchor box formulation. **Hence, proposing to formulate queries of DETR as anchor boxes is novelty itself.** Here, DETR refers to standard Transformer-based DETR models. Most DETR-like models did not have 4D box representations even after Deformable DETR.
> > > - What we highlight in this paper is the DAB-DETR model. We provide DAB-Deformable-DETR just to show you the difference between our DAB formulation and previous methods. DAB-Deformable-DETR also shows that our DAB formulation is general and can further improve Deformable DETR. We are glad to see that you recognized the difference between the two models and gave us more suggestions to analyze the difference.
> > >
> > > D. Why do we say your analysis is incomplete?
> > > - You are trying to show the equivalency of the initialization between the two formulations in 3.1 (1). But we hope that a neater and simpler formulation is a contribution itself. Moreover, this formulation has clearer semantics and can be boosted by other techniques more easily.
> > > - Based on your comments in 3.1 (2), you might think it is the same to learn a high-dimensional query or obtain a high-dimensional query projected from a 4D box. However, a learned high-dimensional query might contain multi-mode information, while the positional information in a high-dimensional query projected from a box is cleaner. That is: $query1(256D) \to box(4D) \to query2(256D) \neq query1(256D) \to query2(256D)$. There is a similar explanation in Conditional DETR.
> > >
> > >
> > > To bo continued

---

> > > ### Author Response · Authors · 2021-11-27
> > > **Responses to your further concerns (2/4)**
> > >
> > > ### Responses to your concerns item-by-item. (continued)
> > > 3 . Ablations for positional query embedding. (continued) \
> > > D. Why do we say your analysis is incomplete? (continued)
> > > - You might try to show the equivalency of the two formulations, but our dynamic anchor box formulation is in some aspects irreplaceable compared with the original representation.  Beyond the three differences you mentioned, in our model, **the positional information passed layer-by-layer is the dynamic anchor box coordinates only**, or say four floats for each query, which cannot be imitated no matter how to modify the model with original query formulation. This is because if you want to ensure the information passed between layers is box coordinates, our formulation is the final choice.
> > >
> > > E. We appreciate your suggestions on the comparison of DAB-Deformable-DETR and Deformable DETR. You have provided great ideas for ablation experiments. We believe these experiments can give us a deeper understanding of our DAB-Deformable-DETR model. We modified our codes accordingly and ran the two suggested ablations. As these experiments are time-consuming, it took us some time to wait for the results. The results are shown in Table R3.1 and Table R3.2 below.
> > >
> > > | Model                  | epoch | AP       | AP$_{50}$ | AP$_{75}$ | AP$_S$ | AP$_M$ | AP$_L$ |
> > > | ---------------------- | ----- | -------- | ----- | ----- | ---- | ---- | ---- |
> > > | DAB-DETR               | 50    | **42.2** | 63.1  | 44.7  | 21.5 | 45.7 | 60.3 |
> > > | DAB-DETR +rm sine func | 50    | 42.1     | 62.5  | 44.7  | 20.9 | 45.9 | 61.2 |
> > >
> > > Table R3.1: comparison of the results between DAB-DETR and DAB-DETR without sine function
> > >
> > > | Model                             | epoch | AP       | AP$_{50}$ | AP$_{75}$ | AP$_S$ | AP$_M$ | AP$_L$ |
> > > | --------------------------------- | ----- | -------- | ----- | ----- | ---- | ---- | ---- |
> > > | DAB-Deformable-DETR               | 50    | **46.8** | 66.0  | 50.4  | 29.1 | 49.8 | 62.3 |
> > > | DAB-Deformable-DETR +rm sine func | 50    | 46.4     | 65.6  | 50.1  | 28.3 | 49.5 | 62.4 |
> > > | Deformable DETR + query update    | 50    | 46.3     | 65.6  | 50.1  | 29.3 | 49.2 | 62.0 |
> > > | Deformable DETR                   | 50    | 46.3     | 65.3  | 50.2  | 28.6 | 49.3 | 62.1 |
> > >
> > > Table R3.2: comparison of the results of some Deformable DETR variants
> > >
> > > F. Impact of the sine function.\
> > > The notion `rm sine func` means we remove the sine function for box projection as suggested. The results in the two tables are different. The results in Table R3.1 show that the impact of the sine function is marginal. However, the results in Table R3.2 (row 1 and row 2) show that it contributes to the final results. The difference might come from our implementation or the special deformable attention mechanism as it does not need to compare the similarity between positional queries and positional keys. Also, as the traditional DETR does not need boxes, our formulation provides a choice to boost the performance with a sine function, which is a strength of our formulation.
> > >
> > > G. Impact of the query update\
> > > For `Deformable DETR + query update` models, we simply project the Transformer decoder outputs to residual positional queries and then add the residual positional queries to original positional queries to make up new positional queries. The results in row 3 and row 4 of Table R3.2 show that there is no significant improvement in the performance after adding iterative update of high-dimensional features. As the semantics of the positional queries is unclear, we do not know what is a good update strategy. On the contrary, iterative anchor box update is much easier as anchor box residual can be directly supervised by the ground truth boxes from the auxiliary loss added to each decoder layer.
> > >
> > > H. The title of this concern is “Is the explicit query positional embedding formulation better than the query positional embedding formulation with implicit location information, or does the performance gain only come from the dynamic update of the query positional embedding?” Yes, we believe the explicit query positional embedding formulation is the key to enable many algorithm designs including the dynamic update of the query positional embedding as we just explained in the previous paragraph, as well as the W&H-modulated attention. This might also explain why Deformable DETR needs to maintain two paths: one path for iterative box update and the other for query positional embedding which are kept the same for all layers (please see Figure 8(e) in our revised paper). It is unclear or nontrivial to update a high-dimensional query positional embedding layer by layer. The results above at least provide evidence that we cannot obtain a gain with dynamic update under the original formulation.
> > >
> > > To be continued

---

> > > ### Author Response · Authors · 2021-11-27
> > > **Responses to your further concerns (3/4)**
> > >
> > > ### Responses to your concerns item-by-item. (continued)
> > > 3 . Ablations for positional query embedding. (continued) \
> > > I. We thank the reviewer for your valuable suggestions. We will add these experiments to the appendix as ablations. We think these questions deserve further study and we will pay more time for them. However, we still want to mention that **all these experiments are good ablations for our DAB-Deformable-DETR, but not evidence to conclude our contribution to one or two specific modules**, as we said in B, C, and D.
> > >
> > > J. The parameter numbers in Table 5 are not wrong. More specifically, the parameter number of Deformable DETR+ (open source) is 46.93M, while the parameter number of DAB-Deformable-DETR(Ours) is 47.18M. The additional parameters introduced are marginal.
> > >
> > > 4 . Novelty of our paper.
> > >
> > > 4 .1 Novelty of dynamic anchor boxes
> > > - We could not agree with your summary of our contributions in the first sentence. What we highlight is the dynamic anchor box formulation other than the iterative update of queries. As we just discussed in the above response, the formulation of dynamic anchor boxes as queries (this is also our paper title) is the key to enabling many algorithm designs, including the iterative update of queries in one path rather than two paths as in both Anchor DETR and Deformable DETR (See Figure 8 (d,e)). **This is the first time the queries of DETR-like models are formulated as dynamic anchor boxes, and we prove many strengths of the formulation**. The update of anchor boxes is one of the advantages under our formulation. Beyond the formulation, the Width&Height-modulated attention is still novel, which will be clarified below.
> > > - We think it’s not fair to summarize contributes by Table 1 only. A better solution for the same target is a progress as well. Also, our dynamic anchor box formulation is different from all DETR-like models as shown in Figure 8.
> > >
> > > 4 .2 Novelty of Width&Height Modulated Attention.
> > > - It is worth noting that our design is a Width&Height-Modulated Attention, while SMCA leverages a modulated Gaussian Kernel.
> > > - We have different objectives: We want to directly modulate the cross-attention map, while SMCA aims to modulate a Gaussian Kernel which is added to the attention map. We apply it on the cross-attention map directly, while SMCA modulates a Gaussian Kernel and then adds them to attention maps.
> > > - We have different formulations and implementations.
> > > - Our design is neater as SMCA has an additional hyperparameter ($\beta$ in Eq. 4 of the SMCA paper).
> > > - We think the only similarity between our Width&Height-modulated attention and SMCA is a similar goal of leveraging the width and height information.
> > >
> > > 4 .3 4D boxes in other DETR-like models.
> > > - We are sorry we have not provided enough explanation in the first response. We will discuss the sentence here.
> > > - You may be taking this sentence the wrong way. **We discuss the necessity of 4D boxes for Deformable DETR in this sentence, while you discuss the feasibility of the iterative update of 4D boxes in other models.** We conclude that the use of 4D boxes is nontrivial for other DETR-like models, while you lead to the conclusion that we can update 4D boxes layer-by-layer for any model. We agree that we can update 4D boxes if you have had them. However, it is not a trivial solution to use 4D boxes in other DETR-like models if we do not use deformable attention or SMCA.
> > > - As deformable DETR needs reference points/boxes, it inevitably needs to generate them from queries. However, there is no such demand in models using the vanilla Transformer attention. What they need are high-dimensional queries as input for attention, but not boxes, hence follow-up works rarely use such designs. That is the reason we said it coupled with the deformable attention mechanism.
> > > - **The Sparse RCNN is not a good example to use against us, but an example to support us.** Because it learns 4D boxes directly as well. It is not a surprise that Sparse RCNN uses 4D boxes as anchor boxes are born with Faster-RCNN. Again, we want to emphasize that it is nontrivial to use 4D boxes as queries in DETR-like models. Here, queries are learnable positional queries. While Deformable DETR has the concept of 4D boxes, we do not regard it as a **complete** design of using 4D boxes as queries because it still keeps a separate path for positional queries which are the same for all layers.
> > > - We want to share our path to the idea of learning anchor boxes directly. In the beginning, we wanted to update the high-dimensional queries layer by layer as you suggested. However, the semantics of queries are unclear, which makes the update unexplainable and difficult to operate. As the 4D box update is more direct, we decided to formulate queries as anchor boxes. This formulation is not a flash of inspiration, but the best choice made after our in-depth thinking and analysis.
> > >
> > > To be continued

---

> > > ### Author Response · Authors · 2021-11-27
> > > **Responses to your further concerns (4/4)**
> > >
> > > ### Responses to your concerns item-by-item. (continued)
> > > 5. Writing suggestions in Sec. 3
> > > - Thank you for recognizing our experiments in Sec. 3.
> > > - Thank you for pointing out where our writing is not clear. As we focused on the comparison between self-attention and cross-attention only, we omitted the FFN modules for simplicity. As both the encoder layers and the decoder layers have FFN modules, the FFN modules are not the key concerns during optimizations.
> > > - We will modify the sentence in the next version. The new sentence will be like: “As the decoder embeddings are initialized as 0, they are projected to the same space as the image features after the first cross-attention module. After that, they will go through a similar process in decoder layers as the image features in encoder layers. Hence the root cause is likely due to the learnable queries.”
> > >
> > > ### Summary:
> > > Thanks for your careful reading and valuable suggestions. Your questions inspired and motivated us a lot to improve our article. We hope you will agree more with our answers. Please feel free to share your further concerns.

---

> > > > ### Comment · Reviewer_zkuH · 2021-11-29
> > > > **Most of my concerns have been addressed.**
> > > >
> > > > It's not easy to present this dynamic anchor boxes idea given the technical similarity with the existing method and the difficulty of interpreting query positional embeddings. But the authors managed to convince me the novelty of this idea. I appreciate the authors' efforts and patience to address my concerns.
> > > > To summarize, this paper proposes a dynamic anchor boxes formulation for the positional queries in detection transformers. The DAB-DETR itself as an object detector is not very attractive to me, but I'm looking forward to seeing the dynamic anchor boxes formulation work as a plug-in-and-play module for existing detection transformers. Thus I'm happy to lift my score.
> > > >
> > > > Additionally, I still have a minor question. Why the parameter number of Deformable DETR+ (open source) is 47M?  The number of parameters in the original Deformable DETR and Deformable DETR+ are 40M and 41M, respectively. Where do the additional parameters come from?

---

> > > > > ### Author Response · Authors · 2021-11-29
> > > > > **Thanks for lifting the score**
> > > > >
> > > > > Thanks for lifting the score. Your comments and suggestions are of great help for us to make this paper clearer. We also hope the neat, general, and effective formulation can be used widely for DETR-like models.
> > > > >
> > > > > To find out the reason for the difference of parameter numbers, we compared the configs of our experiments with the official implementation. We found the parameter `dim_feedforward` is different. The `dim_feedforward` was set to 1024 in the official setting of Deformable DETR, while we ran the experiments with `dim_feedforward=2048` as vanilla DETR. Thanks for your point out. We will rerun these experiments and modify the table accordingly in the paper.

---

### Official Review · Reviewer_H1Mp · 2021-11-02

**Correctness:** 3
**Technical Novelty And Significance:** 3
**Empirical Novelty And Significance:** 3
**Recommendation:** 8
**Confidence:** 4

**Main Review:**

Strengths:
+ This paper is well organized, with clear comparison to prior works in tables and graphs, good problem analysis on the role of queries in DETR and conditional DETR, and informative figures.
+ Experiments demonstrate the advantage of dynamic anchor queries over previous implicit and explicit query formulations including original DETR, conditional DETR and anchor DETR.
+ The proposed query formulation introduces marginal computational overhead compared to other formulations in terms of GFLOPs.

Weaknesses:
- It seems the proposed query formulation works well with the Deformable Convolution Network (DCN) backbone but less competitive without the DCN backbone. For example, DAB-DETR-DC5-R50 outperforms Anchor DETR-DC5-R50 by 1.5 AP but DAB-DETR- R50 outperforms Anchor DETR-R50 by only 0.5 AP. In the ablation study, all experiments are conducted with the DCN. It would be more comprehensive to provide analysis or insights on why dynamic anchor queries cannot offer significant improvements without DCN.
- GFLOPs is a partial indicator of computational efficiency. The authors might want to report the actual runtime speed (ms per image).
- A minor reminder on citation correctness, e.g. Section 4.3 2nd line DETR should not cite Meng et al., 2021.


**Summary Of The Paper:**

This paper introduces dynamic anchor boxes as a query formulation for detection transformer (DETR). The basic idea is to use explicit positional priors and scale priors from anchor boxes as the queries for training the decoder of DETR. Anchor boxes are also adjusted layer-by-layer to learn the optimal anchor setting. The proposed query formulation provides a better understanding of the query for vision transformer and is proven to outperform previous implicit and explicit query formulations.

**Summary Of The Review:**

This paper has the merits of good performance, clear presentation, and a neat idea. Although more analysis and experiments could further improve the manuscript, overall this is a good paper.

---

> ### Author Response · Authors · 2021-11-16
> **Responses to Reviewer H1Mp (R2)**
>
> Thanks for your encouraging review and practical suggestions. Based on your suggestions, we have improved our paper and a revised version is available now. We will first summarize what we have done this week to address your key concerns and then provide detailed responses for each weakness you have pointed out.
>
> ### Summary of your key concerns and our improvements:
> 1. You suggest we discuss the reason for the larger improvement of models with DC5 settings.
> - Our response: We added a paragraph in the main text and discuss it carefully below.
> 2. You suggest we provide the runtime speed
> - Our response: We have tested it on our machines and provided it in Appendix K Table 9 in the revised version.
> 3. You provide some writing suggestions
> - Our response: We have revised the paper accordingly.
>
> ### Detailed responses to your concerns item-by-item.
> 1. The better performance of query proposed on DC5 setting is an interesting phenomenon. As the feature maps of DC5 settings are larger than that in the standard settings, the traditional query formulation is hard to cover all areas. Our model can help each query focus on a certain region, which may result in better performance especially on larger feature maps.
> 2. Thanks for your suggestions. We have provided the runtime speed of our model and Conditional DETR, which is our direct competitor, in Appendix K Table 9. The results show that our model has a similar runtime time compared with Conditional DETR.
> 3. Thanks for your careful reading and revision suggestions. We have fixed the citation accordingly in the revised paper. We will proofread the draft more carefully as well.
>
> ### Summary:
> Thanks for your encouraging review and valuable suggestions. We believe our general, neat, and simple formulation will be used widely for DETR-like models.

---

### Official Review · Reviewer_iJ5R · 2021-11-02

**Correctness:** 2
**Technical Novelty And Significance:** 3
**Empirical Novelty And Significance:** 3
**Recommendation:** 6
**Confidence:** 4

**Main Review:**

## Main strengths:

1) Providing detailed diagrams highlighting the differences between the proposed method and the baselines
2) Performing an experiment explaining to rebut some potential doubts about sources of improvements
3) Visualizing the attention between positional queries and positional keys, justifying the limitations of the previous works and their proposed method's advantage
4) Improving benchmark results over SOTA

## Main Weaknesses:

5) Based on my understanding, the experiment visualized in figure 3, aims to justify that something is wrong with the way positional information is represented and it is not merely an issue with the optimization. The authors should generate training curves of a similar experiment, after making the proposed changes too. It would much better reveal the impact of their changes in the discussed matter and clear the doubt about the reasoning of the experiment, knowing how using "well-trained" queries of their proposed method compare to training them from scratch. furthermore, comparing the described training curves to those in figure 3 would much better illustrate the improvement in rate of convergence which is the main issue targeted by the proposed method. Drawing these curves as a result of multiple runs with an average curve and some measure of confidence could further support the claims based on these types of experiments.
6) Hyperparameter study is limited to temperature and number of decoder layers. Effects of other hyperparameter choices are not clear to the reader. Even if their values are copied from the baseline methods, the proposed changes in the method could alter their behavior. Also, it seems like the provided ablations are compared using the the hyperparameter choices that work best with the proposed configuration. This could mean that those choices are biased towards the proposed method in the ablations. The hyperparameters for each individual configuration should be chosen independent of the others.
7) While Deformable DETR (Zhu et al., 2021) is discussed in the introduction as the last paper in this line of research, its results are not compared in Table 2 and I am not sure the proposed method is necessarily better in terms of AP, and comparative in terms of GFLOPs and number of Parameters.
8) Some of the compared methods had improved results with increased number of training epochs and other configuration of architectures such as backbones. It is not clear how would the proposed method compare to the SOTA in such settings.
9) Qualitative results are quite limited in the paper. From the only provided figure (9), the main visible difference in the last two columns is how the books in the second row of images are detected. A comparison between the proposed method and the baselines highlighting a few examples that contribute to the improvements of the quantitative results could improve this aspect of the paper. Knowing the SOTA on this dataset is still under 50% in AP, visualizing some failure cases and trying to explain why the method failed could further improve the work.

#### Minor points:
10) There is also a paper called " Dynamic DETR: End-to-End Object Detection With Dynamic Attention" from ICCV'21 which better be included in the literature review and compared experiments of this work.
11) Knowing the amount of GPU memory used could be important for reproducing the results.
12) Only releasing the code would not suffice to support the statement on reproducibility. The mentioned points on hyperparameter studies and providing statistical analysis on the results rather than results from a single run, and providing details on the computational infrastructure used could further support this statement.

**Summary Of The Paper:**

The paper addresses the problem of slow convergence in training of object detection transformer. The main contribution is to use trainable anchor boxes (4D) in a layer-by-layer manner. They also motivate their contribution by visualizing positional attentions and explaining the advantage of their proposed method in adapting to scale ratio of objects. They further propose to change the size of the positional attention by an additional temperature parameter, which they choose empirically for the dataset. They perform experiments to justify aspects of their claims, and provide ablations to prove their results being improved over the SOTA on the COCO '17 validation set.

**Summary Of The Review:**

The paper seems to be a step in the right direction to address the issues with detection transformers. The provided figures of architectures are well illustrative and helps readers to understand the differences. The performed experiments to some extent justify the decisions made in the design choices. However, there are missing complementary experiments that I feel are required to fully explain the results. The empirical approaches are lacking as also described in the main review which damages the credibility of some of the claims.
Therefore, I am leaning towards a reject for this version of the work. But I do feel like the paper has the potential to be a great submission in a future revision, assuming the points of concerns are covered with additional experiments. I have provided more details on how to improve the paper. I look forward to see how the authors respond and address my concerns before I finalize my decision.

Edit: I thank the authors for responding to my comments in details.
I am glad to see that the authors found some of my comments useful and took steps to further improve their work.
Some of my main concerns are resolved (5 and 9), while some other concerns are only partially addressed and they still remain a concern in my opinion (6, 7, and 8).

- The authors have not provided even a hint of performing a somewhat systematic/objective hyperparameter search to convince the readers that the hyperparameters are not biased towards the proposed method in the ablation studies such as in table 3.
- Some improved variants of the Deformable DETR paper including the two-stage Deformable DETR and results from the table 3 in that paper, seem to be still missing with appropriate comparisons in the updated draft.
- The authors have provided new results using a couple of alternative architectures using their method which shows potential for further improvements. However, these are not provided side-by-side in comparison to other methods, So that the proposed method can be judged on its capacity for improvement compared to the alternatives discussed in the paper. I do understand that such experiments are time consuming, however, the paper having taken such a comparative approach to motivate its contributions, needs such thorough empirical comparisons to prove its claims on improvements.
- It seems that the concerns of reviewer zkuH regarding contributions being marginally significant or novel, are not sufficiently resolved.

Typos I noticed in the updated draft:
- "Some previous work also has similar analysis and confirmed this.": "work" --> "works"
- "The modulated attentions can be regarded as helping perform soft ROI pooling.": "perform" --> "to perform"

Overall, the updated draft is improved and so I increase the recommendation score to 6.

---

> ### Author Response · Authors · 2021-11-16
> **Responses to Reviewer iJ5R (R1) part 1**
>
> Thanks for your careful reading and helpful suggestions to our paper. We have revised our paper accordingly and the new version is available now. We will first summarize what we have done this week to address your key concerns and then provide detailed responses for each weakness you have pointed out.
>
> ### Summary of your key concerns and our improvements:
> 1. You suggest using multiple runs for experiments in Sec. 3. and providing more discussions about hyperparameters.
> - Our response: We have improved the experiments accordingly and discussed the choice of hyperparameters with some new experiments in Appendix H.
> 2. You suggest using a stronger backbone for better performance.
> - Our response: We have improved our model with some new techniques including multi-scale and stronger backbone like Swin-Transformer and report some stronger results in our detailed responses below.
> 3. You suggest providing more qualitative results.
> - Our response: We provide more qualitative analyses including a comparison of box updates and analysis of failure samples, which are also available in Appendix I and J.
>
> ### Detailed responses to your concerns item-by-item.
> Main Weaknesses:
>
> **5.**
> - We run the experiments in Sec.3 3 times and draw the mean value and 95% confidence interval in our revised paper. See Sec. 3 Fig. 3 (a). However, due to the high cost of each DETR experiment, e.g. we need 8 Nvidia A100 GPUs to run training for about 5 days or 32 Nvidia A100 GPUs to run training for more than one day for each experiment, we don't have enough time to repeat all experiments for 3 times, which will be improved in the next version.
> - We will explain the meaning of "well-trained" queries. We initialize the queries of DETR using the parameters from a well-trained model and will not update them by gradient descent. If DETR queries are hard to learn, "well-trained" queries will help the model converge faster. However, the training curves show that there is no obvious difference between the two classes of models. Hence the learning of queries is not the key problem of the original DETR.
> - We fully agree with your suggestion about experiments. Hence we provide both the AP vs. epoch and the loss vs. epoch curves in the Sec. 3 of the revised paper. The trend of the loss curves also confirms our conclusion.
> - We added a new experiment comparing the original DETR with a simple DETR variant DETR+DAB by adding our proposed dynamic anchor boxes. Note that DETR+DAB is different from our DAB-DETR. We only replace the query formulation in DETR with our proposed dynamic anchor box formulation, without other tricks or improvements. We believe the new experiment can lead to the conclusion that the multi-mode property of the original DETR queries is at least one of the key reasons for slow convergence.
> **6.**
> - We agree that a discussion about the choice of hyperparameters in the experiments is important. However, as there are many hyperparameters, we only tried a small set of them. Most changes of hyperparameters have no significant improvement in the final results. Interestingly, we found the batch size and the initialization of (x, y) of anchor box centers have impacts on the performance. The model can achieve the best performance with a batch size 16. We provide the results of our DAB-DETR with random initialization of anchor box center (x, y) in appendix H of our revised paper. We think these are interesting observations and deserve deeper investigation in the future.
> - We have tried to find better results in the ablation study. However, we found the best config of our full-featured model is already a good setting for ablation models. We will do more experiments for a more convincing comparison.
> **7.** Actually, we compared Deformable DETR with our model in the main table (Table 2) in the first version of our paper (see the second line of the second block in Table 2). Although they use the multi-scale technique, our models are still better than the results reported in their paper.
>
> To be continued.

---

> ### Author Response · Authors · 2021-11-16
> **Responses to Reviewer iJ5R (R1) part 2**
>
> Continued to responses part 1.
>
> ### Detailed responses to your concerns item-by-item. (continued)
> Main Weaknesses:
>
> **8.** Thank you for your advice. We have tested two larger models these days. The results are shown in the table below. The model with “MS” means we leverage the multi-scale technique and the model with “SwinL” means we replace the original backbone with Swin-Transformer. These experiments only show the preliminary potential of our models. We have noted that the state-of-the-art models usually use an even stronger backbone (like Swin-Transformer with feature pyramid) and a larger dataset (like Object365) to achieve better performance. Later we will follow these practices to further improve our model and fully test its potential for a new SOTA.
>
> | model                | AP   | AP$_{50}$ | AP$_{75}$ | AP$_{S}$ | AP$_{M}$ | AP$_{L}$ |
> | -------------------- | ---- | ----- | ----- | ---- | ---- | ---- |
> | DAB-DETR-R50-DC5*    | 45.7 | 66.2  | 49.0  | 26.1 | 49.4 | 63.1 |
> | DAB-DETR-R50-DC5*-MS | 46.9 | 66.0  | 50.8  | 30.1 | 50.4 | 62.5 |
> | DAB-DETR-SwinL       | 52.9 | 74.3  | 57.1  | 33.2 | 57.9 | 72.6 |
>
> **9.** Thanks for your constructive suggestions. We have provided two qualitative results in the revised paper: a comparison of the changes of the predicted results between Conditional DETR and our DAB-DETR in Appendix I, and an analysis of failure cases in Appendix J. We hope the two qualitative results can help us better understand and improve our models.
>
> **10.**
> - Thanks for your advice. As the Dynamic DETR paper was not published until the ICCV conference which was held after the ICLR submission deadline, we couldn't make a comparison with it in our submission. In the revised version, we have added a discussion about Dynamic DETR and our DAB-DETR in Appendix C. Dynamic DETR uses multi-scale features, which are further enhanced by a dynamic encoder technique called Dynamic Head. Their backbone + dynamic encoder is a much stronger backbone than our backbone. But our simple multi-scale version still achieves a promising performance compared with Dynamic DETR (ours 46.9 on val v.s. theirs 47.0 on test).
> - As Dynamic DETR is not open-sourced, some implementation details are not fully clear. A detailed discussion of Dynamic DETR is available in Appendix C paragraph 4.
>
> **11.** We provide the GPU resource needed during training in Table 4 of Appendix A. We train all models with Nvidia A100 GPUs. The description of the GPU we used is added to Appendix A as well.
>
> **12.** Thanks for your suggestions and we change the statement on reproducibility accordingly. We will release our code, all pretrained models and settings, and other materials needed to reproduce our experiments. Your suggestions are of great help for us to improve this article.
>
> ### Summary:
> Thank you for your constructive comments and suggestions. They are very helpful for us to improve the article. We did a lot of new experiments and tailored the article accordingly.

---

> ### Author Response · Authors · 2021-11-29
> **Thanks for lifting the score**
>
> Thank you for your positive recommendation. We will modify our paper accordingly in the next version. More specifically, we will add hyperparameter searching for ablation experiments, add comparisons with variants of Deformable DETR properly in Table 3, and add experiments with competitors of improved techniques. We will fix the typos and proofread the paper more carefully as well. Moreover, we are glad to see that most concerns of reviewer zkuH have been addressed. We will modify the paper to make it clearer.

---

### Author Response · Authors · 2021-11-16
**Summary of Paper Revision**

We appreciate all the reviewers for their valuable and constructive comments. We have revised our paper accordingly and summarized the main improvements below:

1. We added a new experiment in Sec. 3 to verify our assumption.
2. We run all experiments in Sec. 3 3 times and plotted the mean value and 95% confidence interval in Fig. 3.
3. We added the GPU usage in Sec. A.
4. We added a detailed comparison of DETR-like models in Sec. B.
5. We designed a DAB-Deformable-DETR model with faster convergence and better performance in Sec. C.
6. We added a new experiment of the initialization of (x,y) in Sec. H.
7. We added two qualitative analyses in Sec. I and Sec. J.
8. We reported the runtime of our model and competitors in Sec. K.

---

### Decision · Program_Chairs · 2022-01-20

**Decision:**

Accept (Poster)

**Comment:**

Somewhat borderline paper given the scores, but leaning on the side of accepting mostly because the positive (and weak positive) reviews are a little more persuasive. The negative review is a bit of an outlier; the main issues raised in the negative review are that the novelty is on the lower side or otherwise that the work is incremental. These complaints are largely not shared by the other reviewers, and furthermore seem not like deal-breakers. Still a borderline paper, but fairly safe to accept.